# Comparative quantum-classical dynamics of natural and synthetic molecular rotors show how vibrational synchronization modulates the photoisomerization quantum efficiency

Alejandro Blanco-Gonzalez [1], Madushanka Manathunga [1,3], Xuchun Yang [1] & Massimo Olivucci [1,2] ✉

We use quantum-classical trajectories to investigate the origin of the different photoisomerization quantum efficiency observed in the dim-light visual pigment Rhodopsin and in the light-driven biomimetic molecular rotor *para*-methoxy N-methyl indanylidene-pyrrolinium (MeO-NAIP) in methanol. Our results reveal that effective light-energy conversion requires, in general, an auxiliary molecular vibration (called promoter) that does not correspond to the rotary motion but synchronizes with it at specific times. They also reveal that Nature has designed Rhodopsin to exploit two mechanisms working in a vibrationally coherent regime. The first uses a wag promoter to ensure that ca. 75% of the absorbed photons lead to unidirectional rotations. The second mechanism ensures that the same process is fast enough to avoid directional randomization. It is found that MeO-NAIP in methanol is incapable of exploiting the above mechanisms resulting into a 50% quantum efficiency loss. However, when the solvent is removed, MeO-NAIP rotation is predicted to synchronize with a ring-inversion promoter leading to a 30% increase in quantum efficiency and, therefore, biomimetic behavior.

The design of molecules capable of converting light energy into mechanical motion with enhanced efficiency is part of contemporary research on molecular devices. One research line focuses on N-methyl indanylidene pyrrolinium (NAIPs) mimic of the visual pigment rhodopsin (Rh). Indeed, NAIPs are light-driven molecular switches that replicate different aspects of the photoisomerization of the 11-*cis* retinal chromophore (rPSB11) of Rh[1–3]. As shown in Fig. 1a, the canonical switch *para*-methoxy-NAIP mimic (MeO-NAIP) features, similar to rPSB11, a cationic Schiff base function and a reactive C1−C2(R1)=C3(R4)−C4 ethylenic moiety. Transient absorption spectroscopy (TAS) has demonstrated that, like rPSB11 in Rh, MeO-NAIP in methanol undergoes sub-picosecond[4]

double-bond photoisomerization corresponding to the rotary power-stroke of molecular motors[5,6]. Remarkably, additional TAS studies have shown that both systems display coherent (i.e. phased) nuclear motion, suggesting a common topography of their excited ($S_1$) and ground ($S_0$) state potential energy surfaces (PESs)[7–10]. Consistently, computational studies have shown that, in both Rh and MeO-NAIP, the reaction occurs via relaxation along a barrierless $S_1$ PES, eventually leading to decay to $S_0$ in the region of an $S_1/S_0$ conical intersection (CoIn). These studies also showed that, in both systems, $S_1$ corresponds to a π−π* excitation dominated by a charge-transfer ($1B_u$) character with respect to the covalent/diradical ($1A_g$) character of $S_0$.

[1]Department of Chemistry and Center for Photochemical Sciences, Bowling Green State University, Bowling Green, OH 43403, USA. [2]Department of Biotechnology, Chemistry and Pharmacy, University of Siena, I-53100 Siena, Italy. [3]Present address: Department of Chemistry and Department of Biochemistry and Molecular Biology, Michigan State University, East Lansing, MI 48824, USA. ✉e-mail: massimo.olivucci@unisi.it

**Fig. 1 | Structures and reactivity of systems under comparison. a** rPSB11 and MeO-NAIP structures and their common ethylenic framework and definition of the skeletal (C1, C4) and ethylenic (R1, R4) substituents. Notice the consistent out-of-plane counterclockwise (CCW) deformation in the two systems. **b** Coordinates driving the photoinduced double bond isomerization: the skeletal bond-length-alternation (BLA), the double-bond twisting of the ethylenic fragment ($\alpha$), dihedral describing the wag ($\beta$) of the ethylenic substituents. **c** Definition of $\tau$, $\alpha$, and $\delta_{op}$, where $\tau$ is the dihedral defining the overlap between the $p$-orbitals (represented by the axis $a$ and $a'$ at C2 and C3, and $\delta_{op}$ is a convenient function of $\beta$ describing the substituent wag relative to the molecule conjugated framework. $0.5\delta_{op}$ is also

defined at the lower right corner of the panel. **d** Schematic illustration of the potential energy surfaces (PESs) of both the ground state ($S_0$) and the excited state ($S_1$) (in solid black line). The trajectory represented through the evolution of double-bond isomerization coordinates BLA, $\alpha$, and $\delta_{op}$ (in dashed lines). The wavelength of maximum absorption ($\lambda_{max}$) defines the electronic transition. The red and blue circles represent reactive and unreactive decay points, respectively. **e** Relationship between orbital overlap, $d\tau/dt$ phase, and reactivity. A reactive event can be associated with the $\pi$-bond breaking ($d\tau/dt < 0$), and the unreactive event is associated with the $\pi$-bond making ($d\tau/dt > 0$), defined by the $p$-orbitals ($a$ and $a'$) overlap at decay.

In the past, the above NAIP's biomimetic features have been associated with an enhanced light-to-mechanical energy conversion[1–3]. However, similar to non-biomimetic crowded-alkene-based motors[11], NAIPs in solution only achieve a ca. 20%[1,2,12] isomerization quantum efficiency ($\Phi^{iso}$) that is far from the ca. 70% value[13,14] measured for the rPSB11 isomerization in Rh. While such a result is disappointing, it provides a unique opportunity for building a theoretical framework linking molecular structure and $\Phi^{iso}$ variations. The present work attempts to establish such a link that is not arguably of basic relevance for the field of photochemistry.

Three modes are implicated in the $S_1$ isomerization coordinates of rPSB11 and MeO-NAIP (see Fig. 1b). These are defined as the skeletal bond-length-alternation (BLA) stretching mode incorporating the C2–C3 ethylenic bond expansion, the C1–C2–C3–C4 dihedral angle describing the double-bond twisting ($\alpha$) of the ethylenic fragment and the R1–C2–C3–R4 dihedral describing the wag ($\beta$) of the ethylenic substituents relative to the conjugated backbone (see Fig. 1b). The $\alpha$

and $\beta$ modes also provide information on the progression of $\pi$-bond breaking and making. In fact, as shown in Fig. 1c, they define $\tau$, a dihedral angle proportional to the overlap between the $p$-orbitals at C2 and C3 (i.e., these are represented by the axis $a$ and $a'$ in Fig. 1c and d), which form the ethylenic $\pi$ and $\pi^*$ molecular orbitals. As schematically shown by the Newman projection of Fig. 1c, $\tau = 0°$ and $\tau = -180°$ are associated with the large overlap characterizing the *cis*-reactant and *trans*-product, respectively. $\tau = -90°$ is instead associated with the null overlap of CoIn points (see Fig. 1d). Notice that, here, we only consider one of the two mirror-image conformers of NAIP. As shown in Fig. 1a, the selected conformer features a C1–C2(R1)=C3(R4)–C4 pre-twisting mimicking the C10–C11(H)=C12(H)–C13 pre-twisting of rPSB11 in Rh. Since such pre-twisting biases the rotary power-stroke in the counterclockwise (CCW) direction (i.e. towards decreasing values of $\tau$), Rh and NAIP are seen as light-driven unidirectional rotors.

Our theoretical framework assumes that the velocity of $\tau$ ($d\tau/dt$) is the critical quantity for predicting the evolution of $\pi$-bond making

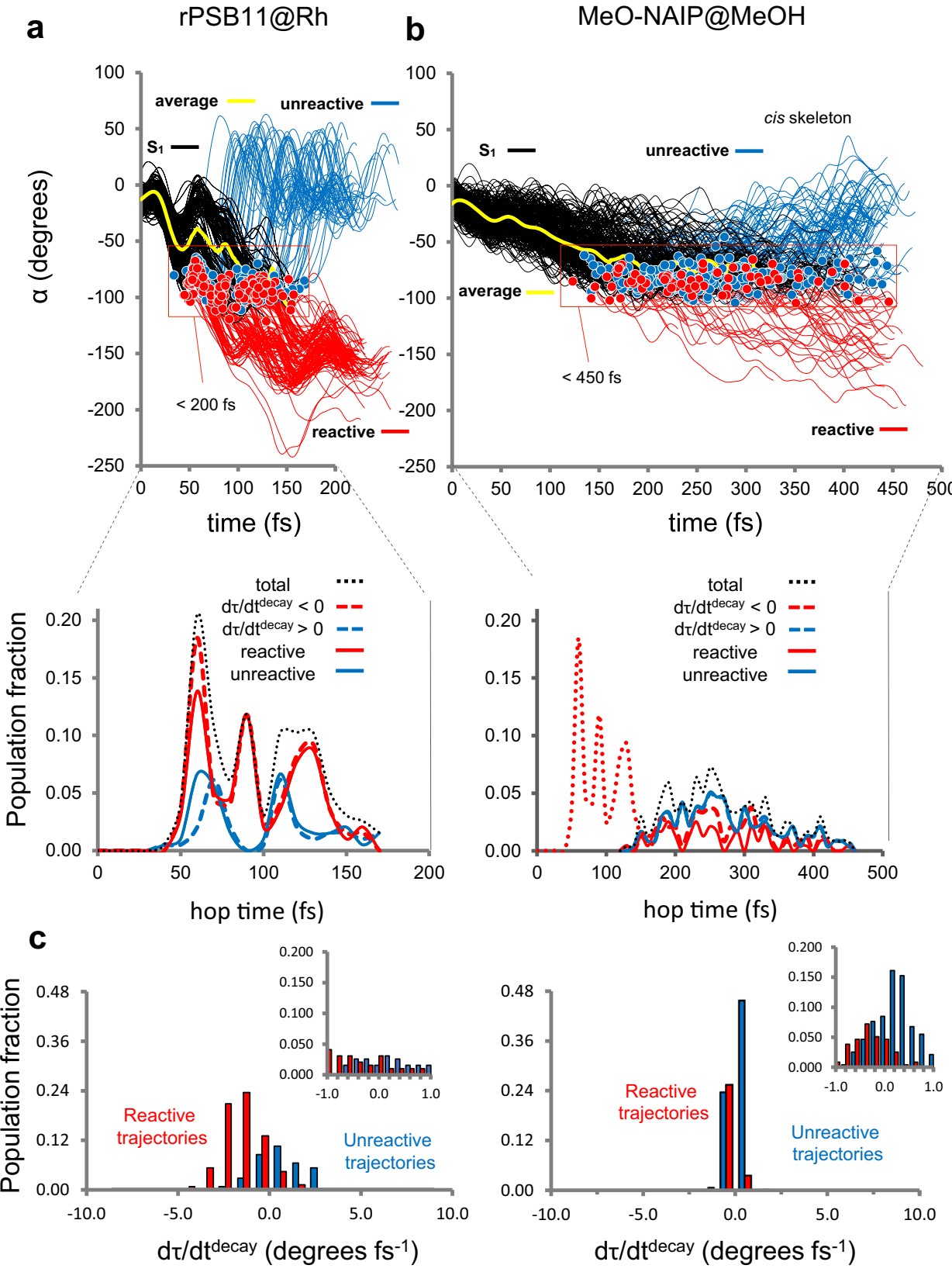

process at the trajectory level. As previously reported for Rh[15–17], it is the phase (sign) of such a velocity that determines the reactivity. Indeed, the analysis of entire sets of quantum-classical trajectories describing the rPSB11 isomerization in Rh has shown that when $d\tau/dt$ at the point of $S_1 \rightarrow S_0$ decay (from now on $d\tau/dt^{decay}$) < 0, the trajectory leads, within a certain tolerance, to the all-trans-product. In contrast,

when $d\tau/dt^{decay}$ > 0, the trajectory re-generates the 11-cis-reactant (see Fig. 1d). It is apparent that, when applied to the motion of the entire molecular population, such reactivity condition creates a link between the $d\tau/dt^{decay}$ distribution and $\Phi^{iso}$ defined as the fraction of reactive trajectories. Furthermore, the fact that $\tau$ is a function of $\alpha$ and $\delta_{op}$ (a convenient function of $\beta$, see Fig. 1c) provides a connection between

**Fig. 2 | Comparison of the photoisomerization dynamics of rPSB11@Rh and MeO-NAIP@MeOH. a** Top: $\alpha$ evolution of rPSB11@Rh. The black lines represent the population motion along $S_1$ before the decay point (represented by red and blue circles), and the yellow curves follow the average value. The red lines indicate the trajectories reaching the photoproduct on $S_0$ (reactive trajectories), and the blue lines are the trajectories reaching the original reactant (unreactive trajectories). Bottom: Oscillatory behavior in the reactive trans-product formation. **b** Same data for MeO-NAIP@MeOH. The red dotted line is the total population of rPSB11@Rh

seen in (**a**) compared to the MeO-NAIP@MeOH time scale. **c** $\tau$ velocity fraction at the decay time (hop time) for the entire population. Reactive trajectories are in red bars, and unreactive trajectories are in blue bars. Most reactive trajectories (ca. 90% in both cases) have $d\tau/dt^{decay} < 0$, while a minor number (ca. 10% in both cases) of trajectories with low positive amplitude $d\tau/dt^{decay} > 0$ are found to be reactive. The bins are 1.0 wide starting from -10.0. The insets show a finer distribution of the bins around the 0.0 value.

$\Phi^{iso}$ and the geometry of rPSB11. Such a connection has allowed probing the $d\tau/dt^{decay} < 0$ reactivity condition experimentally[15–17], showing that deuterium substitution on the H-C11=C12-H moiety of rPSB11 (i.e., a perturbation of $\delta_{op}$) modulates the $\Phi^{iso}$ value[15].

The studies revised above hold that the $d\tau/dt^{decay}$ phase and amplitude must depend on the synchronization of $\alpha$ with $\delta_{op}$ during their progression towards the $S_1/S_0$ intersection space ($IS_{S1/S0}$, i.e. the 3N-8 dimensional subspace of the nuclear Cartesian coordinates N, formed by all possible CoIn points). At the level of single-trajectory contributions, $\Phi^{iso}$ can be discussed using the model illustrated in Fig. 1d, e. Light excitation projects a molecule to the $S_1$ state. The $S_1$ PES then accelerates the molecule along BLA, $\alpha$, and $\delta_{op}$. It is apparent from the relationship in Fig. 1c that the phase relationship between $d\alpha/dt^{decay}$ and $d\delta_{op}/dt^{decay}$, resulting from the synchronization of $\alpha$ and $\delta_{op}$ motions, determines $d\tau/dt^{decay}$ and, therefore, if the trajectory is reactive/successful (red circle) or unreactive/unsuccessful (blue circle).

Below we use the $\tau$, $\alpha$, and $\delta_{op}$ velocities extracted from hundreds of quantum-classical trajectories (we use Tully's fewest switches surface hopping method[18] with the Persico–Granucci decoherence correction[19]) to uncover the origin of the different $\Phi^{iso}$ values observed for rPSB11 and MeO-NAIP in different environments. It is shown that the resulting mechanistic interpretation leads to a generalization of the theoretical framework based on the $d\tau/dt^{decay} < 0$ reactivity condition that also includes MeO-NAIP. Such a generalization implies that such a condition is only valid when the absolute magnitude of $d\tau/dt^{decay}$ overcomes a certain threshold. As discussed below, this is qualitatively in line with the canonical Landau–Zener model.

More specifically, (i) we find that the $d\tau/dt^{decay} < 0$ condition is a sufficient condition only when the $d\tau/dt^{decay}$ amplitude is $\lesssim -1$ degree/fs and that such amplitude is modulated by a "promoter mode" affecting $\delta_{op}$, not the reaction coordinate $\alpha$; (ii) the promoter mode is different and has a different frequency in natural and synthetic homolog; (iii) the effect of the promoter can be quenched by molecular environments that modify the rotor electronic structure. We argue that (i)–(iii) are novel findings defining an original mechanism that, in principle, can be used to design photochemical systems with enhanced $\Phi^{iso}$ values.

## Results and discussion

Our research is based on quantum-mechanics/molecular-mechanics (QM/MM) models of rPSB11[13,14] embedded in the Rh (opsin) cavity (rPSB11@Rh) and the synthetic MeO-NAIP[1–3] embedded in the solvent (methanol) cavity (MeO-NAIP@MeOH).

### Reactive and unreactive trajectories

Figure 2a, b display the evolution of $\alpha$ for the CCW isomerization of 200 trajectories for both rPSB11@Rh and MeO-NAIP@MeOH (see Supplementary Data 1 and 3 for the corresponding numerical data). Following the schematic representation of Fig. 1d, e, each trajectory is classified as reactive (marked by a red circle) and unreactive (marked by a blue circle) according to the configuration reached after the hop. As expected, all hop points have $\alpha$ values close to $-90°$. Consistently with the experimental observations, the rPSB11@Rh population decays within 200 fs while MeO-NAIP@MeOH is slower and requires not <450 fs to decay. The diversity in decay times is attributed to different $d\alpha/dt$ amplitudes. In fact, it is evident from the figure that rPSB11@Rh starts to decay as early as 30 fs while MeO-NAIP@MeOH reaches the

decay region on a longer 150 fs time scale. The time length between the beginning and the end of the decay process is also significantly different. In rPSB11@Rh, it is completed in ca. 150 fs, while in MeO-NAIP@MeOH takes ~350 fs.

The matching between simulated and observed differences in the spectroscopy and dynamics of rPSB11@Rh and MeO-NAIP@MeOH (see the "Methods" section below for the complete list of observed vs. computed quantities) support the use of the corresponding models in mechanistic studies. For instance, as reported in Table 1, the calculated $\Phi^{iso}$ values (i.e., the fraction of reactive trajectories) are 67% and 30% for rPSB11@Rh and MeO-NAIP@MeOH, respectively, and are comparable to the observed 67%[13,14] and 21%[1,2,12] values.

The bottom parts of Fig. 2a, b show that Rh displays an approximately periodic product formation. This is consistent with $S_1$ coherent nuclear motion (here with "coherent motion" we do not refer to the coherence generated by a laser pulse but, rather, to a phased motion induced by the $S_1$ force field upon excited state relaxation). In fact, the rPSB11@Rh reactive trajectories form 3 or 4 sets (see full red line) separated by a 30–40 fs period. Below we propose that such oscillating reactivity is produced by an oscillation in the $d\tau/dt^{decay}$ sign. In contrast, in MeO-NAIP@MeOH, the product formation process is consistent with a less populated and more structureless set of reactive trajectories. Notice that, in both cases, the unreactive decays are, somehow, less organized. Below, we focus on the mechanism generating reactive trajectories by examining the robustness of the reactivity condition $d\tau/dt^{decay} < 0$.

### $d\tau/dt^{decay} < 0$ is not a sufficient condition for reactivity

The data in Table 1 shows that 93% and 87% of the reactive trajectories of rPSB11@Rh and MeO-NAIP@MeOH are associated with $d\tau/dt^{decay} < 0$, while only 7% and 13%, respectively are associated with $d\tau/dt^{decay} > 0$. If we now accept a lost 15% tolerance for the validity of the reactivity condition $d\tau/dt^{decay} < 0$, we can conclude that this is valid for both systems (the analog relationship $d\tau/dt^{decay} > 0 =$ unreactive is documented in Section IV of the Supplementary Information). Previous work on rPSB11@Rh treated this as a sufficient condition implying that a trajectory with $d\tau/dt^{decay} < 0$ is always reactive[15,20,21]. However, while the simulation of the Rh dynamics complies with a sufficient condition within a certain tolerance (Table 2 shows that 85% of the trajectories with $d\tau/dt^{decay} < 0$ are reactive), this is not true for

**Table 1 | Computed isomerization quantum efficiency ($\Phi^{iso}$) and $d\tau/dt^{decay}$ phase**

| Criteria | Full population[a] | |
|---|---|---|
| | rPSB11@Rh (%) | MeO-NAIP@MeOH (%) |
| Reactive ($\Phi^{iso}$) | 67 [51] | 30 [62] |
| | Reactive population fraction[b] | |
| | rPSB11@Rh (%) | MeO-NAIP@MeOH (%) |
| $d\tau/dt^{decay} < 0$ | 93 [82] | 87 [77] |

The percentages correspond to rPSB11@Rh and MeO-NAIP@MeOH population fractions satisfying the criteria given in the first column. The values in square brackets refer to the isolated chromophores.
[a]% with respect to the full population.
[b]% with respect to the reactive population defined by looking at the $S_0$ configuration reached after the hop.

**Table 2 | Analysis of the full population**

| Criteria | Full population[a] | |
|---|---|---|
| | rPSB11@Rh (%) | MeO-NAIP@MeOH (%) |
| $d\tau/dt^{decay} < 0$ | 73 [63] | 50 [72] |
| | Population fraction with $d\tau/dt^{decay} < 0$[b] | |
| | rPSB11@Rh (%) | MeO-NAIP@MeOH (%) |
| Reactive[c] | 85 [66] | 52 [62] |

The percentages correspond to rPSB11@Rh and MeO-NAIP@MeOH population fractions satisfying the criteria given in the first column. The values in square brackets refer to the isolated chromophores.

[a]% with respect to the full population

[b]% with respect to the $d\tau/dt < 0$ population.

[c]Reactive population defined looking at the $S_O$ configuration reached after the hop.

**Table 3 | Analysis of the populations with $d\tau/dt^{decay} < 0$**

| Criteria | Population fraction with $d\tau/dt^{decay} < 0$[a] | |
|---|---|---|
| | rPSB11@Rh (%) | MeO-NAIP@MeOH (%) |
| $-0.8 < d\tau/dt^{decay} < 0$ | 24 [28] | 90 [75] |
| | Population fraction with $-0.8 < d\tau/dt^{decay} < 0$ | |
| | rPSB11@Rh (%) | MeO-NAIP@MeOH (%) |
| Reactive[b] | 54 [41] | 48 [56] |

The percentages correspond to rPSB11@Rh and MeO-NAIP@MeOH population fractions satisfying the criteria given in the first column. The values in square brackets refer to the isolated chromophores.

[a]% with respect to the full $d\tau/dt^{decay} < 0$ population.

[b]Reactive population defined looking at the $S_O$ configuration reached after the hop.

MeO-NAIP@MeOH. In fact, in such a system only 52% of the trajectories with $d\tau/dt^{decay} < 0$ are reactive. In other words, an almost equal proportion of reactive and unreactive trajectories has $d\tau/dt^{decay} < 0$; therefore $d\tau/dt^{decay} < 0$ cannot be, in general, a sufficient condition for reactivity.

The consequence of the fact that $d\tau/dt^{decay} < 0$ is not a sufficient condition is that the $\Phi^{iso}$ values computed using that criterium are overestimated. As shown in Table 2, the computed values would be 73% and 50% rather than 67% and 30% (see Table 1) provided by the reactive/unreactive statistics. At this point, we recall that $d\tau/dt$ is assumed to represent the $p$-orbital overlap and, therefore, of the π-bond making. Thus, not all CCW π-bond making processes are completed, but many invert directions to regenerate the reactant. It is now necessary to investigate under which circumstances this inversion occurs.

### $d\tau/dt^{decay} < 0$ with a ca. $-1$ degree/fs amplitude is a sufficient condition for reactivity

Figure 2c shows that, for rPSB11@Rh, the unreactive trajectories (blue bars) associated with a $d\tau/dt^{decay} < 0$ have a significantly smaller amplitude (see abscissa between 0.0 and −2.0) than the reactive ones. The same appears to be true for the small population fraction (7%) of reactive trajectories (see Table 1) that have a $d\tau/dt^{decay} > 0$ (see abscissa between 0 and +2.0). Thus, when the absolute $d\tau/dt^{decay}$ amplitude falls below a certain threshold, the reactivity condition weakens, and the direction of decay is randomized. Inspection of the corresponding diagram for MeO-NAIP@MeOH shows that the same conclusion applies to this system but in a more dramatic way. Indeed, in MeO-NAIP@MeOH, the whole population shows $d\tau/dt^{decay}$ with much lower absolute amplitudes (similarly having a small population fraction (13%) of reactive trajectories (see Table 1) with $d\tau/dt^{decay} > 0$).

An analysis of the $d\tau/dt^{decay} < 0$ population was conducted to support the hypothesis that the $d\tau/dt^{decay}$ absolute amplitude provides an additional reactivity condition. In short, $d\tau/dt^{decay} > 0$ is, in general,

valid only when the absolute value of $d\tau/dt^{decay}$ overcomes a certain threshold. Accordingly, we look at a fraction of rPSB11@Rh and MeO-NAIP@MeOH populations with a $d\tau/dt$ value comprised between 0 and −0.8 degrees/fs (see Table 3). It was found that, in both cases, the percentage of trajectories that generated a product was close to 50%.

The reduced (54% see Table 3) photoisomerization efficiency of the rPSB11@Rh population fraction with $d\tau/dt^{decay} < 0$ and amplitude below the threshold contributes to explaining why the computed $\Phi^{iso}$ value (67%) is significantly lower than the fraction of trajectories with $d\tau/dt^{decay} < 0$ (73%). The same reasoning applies to MeO-NAIP@MeOH. In this case, the fraction with $d\tau/dt^{decay} < 0$ and amplitudes below the threshold (90%, see Table 3) have even lower photoisomerization efficiency (48%, see Table 3) and contributes to explain the computed $\Phi^{iso}$ value (30%, see Table 1), again lower than the fraction of the population with $d\tau/dt^{decay} < 0$ (50%, see Table 2).

In conclusion, the percentage of trajectories with $d\tau/dt^{decay} < 0$ determines the $\Phi^{iso}$ value. However, the sign of $d\tau/dt^{decay}$ becomes less significant for trajectories whose $d\tau/dt^{decay}$ amplitude falls below a ca. −1.0 degree/fs threshold. This "velocity-magnitude-dependent" behavior may be rationalized using the Landau−Zener (LZ) model[22,23] that, for the rotors under investigation, would be applied to $\tau$ (i.e., a combination of $\alpha$ and $\delta_{op}$) during the transit across the IS$_{S1/S0}$. Indeed, the LZ model predicts a reactivity proportional to the $d\tau/dt^{decay}$ amplitude. When the $d\tau/dt^{decay}$ amplitude becomes too small, the reactivity is determined by more subtle factors (e.g., the PES slopes). In the following two sections, we discuss how the natural rotor uses shorter population decay times as well as the phase relationship between the geometrical parameters $\alpha$ and $\delta_{op}$ to achieve a larger $d\tau/dt^{decay} < 0$ population fraction with larger $d\tau/dt^{decay} < 0$ amplitudes.

### A promoter mode modulates the $d\tau/dt$ amplitude

As reported above (see Fig. 2a, b), the progression in the skeletal twisting $\alpha$ shows a 30−150 fs increase in the initial decay time (IDT) when moving from rPSB11@Rh to MeO-NAIP@MeOH. The same data show a corresponding 150−350 fs increase in decay time length (DTL), suggesting that the decrease in IDT is associated with an increase in DTL or, equivalently, a decrease in the average $d\alpha/dt$ is associated with a broadening of the range of $d\alpha/dt$ values at decay.

As detailed in Section V of the Supplementary Information, we constructed a basic model of the population dynamics where the average $\alpha$ is expressed as the sum of a monotonic ($\alpha_I$) and an oscillatory ($\alpha_{II}$, same period of $\delta_{op}$) progression towards IS$_{S1/S0}$ (see Fig. 3a). The model indicates that in systems with a short IDT and narrow DTL (e.g. rPSB11@Rh), the $\alpha_{II}$−0.5$\delta_{op}$ velocity promotes the power-stroke that, when synchronized with $d\alpha_I/dt$, periodically enhance the amplitude of $d\tau/dt^{decay} < 0$. We call a promoter any mode enhancing the $\alpha_{II}$−0.5$\delta_{op}$ amplitude. In contrast, in systems with long IDT and, therefore, large DTL corresponding to a broad $S_1$ decay and wide distribution of low amplitude $d\alpha_I/dt^{decay} < 0$ values (e.g., as in MeO-NAIP@MeOH), the promoter is not effective. In the following, we assume that the average −0.5$d\delta_{op}/dt$ and $d\alpha/dt$ are proportional to $d(\alpha_{II}$−0.5$\delta_{op})/dt$ and $d\alpha_I/dt$, respectively, and can be used to discuss the factors responsible for the population with larger $d\tau/dt^{decay} < 0$ population and increased $d\tau/dt^{decay}$ absolute amplitudes displayed by rPSB11@Rh with respect to MeO-NAIP@MeOH. More specifically, the amplitude enhancement must be directly proportional to the velocity of the C2=C3 isomerization mode ($d\alpha/dt$), as well as to the promoter velocity that will increase the −0.5$d\delta_{op}/dt$ absolute amplitude.

The computed initial population progression supports the conclusions above. In Fig. 3b−d, we compare the rPSB11@Rh and MeO-NAIP@MeOH distributions of $d\tau/dt$, $d\alpha/dt$ and $d\delta_{op}/dt$, 1 and 15 fs after photoexcitation. It is apparent that during the initial progression, the natural rotor dramatically increases the $d\tau/dt < 0$ amplitude, yielding a large fraction (see colored area) of the population overcoming the ca. −1.0°/fs threshold in the reactive direction. In contrast, the synthetic

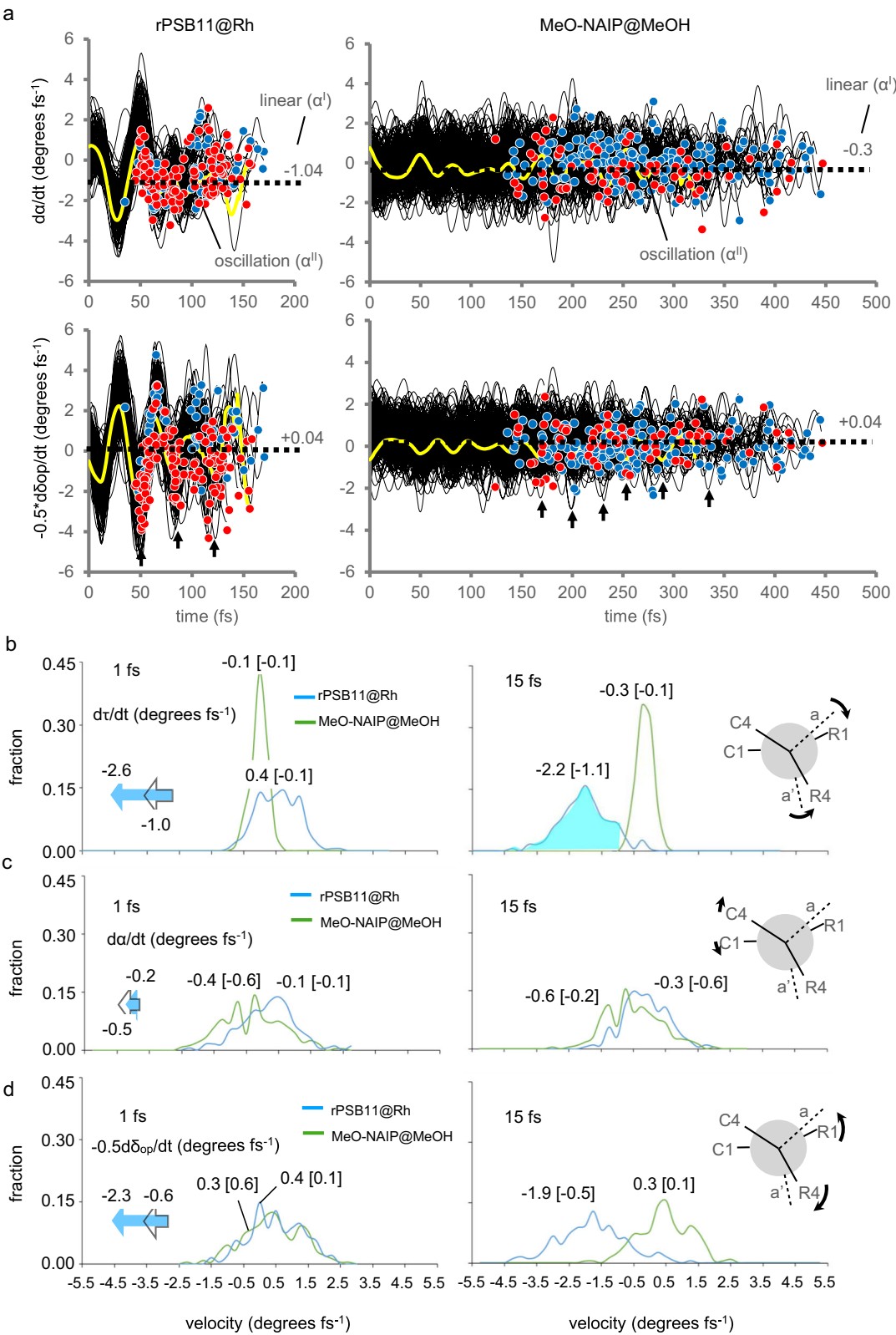

rotor conserves a low initial velocity. These same data indicate that a promoter with a 40-fs period (possibly corresponding to the spring effect proposed in the literature)[24] operates in the natural rotor but not on the synthetic rotor. As shown in Fig. 3c, $d\alpha/dt$ undergoes a modest velocity increase, however Fig. 3d displays a strong velocity increase of $-0.5d\delta_{op}/dt$ due to the wag promoter (i.e. the R1, R2 substituents wag) in rPSB11@Rh but not in MeO-NAIP@MeOH. We thus hypothesize that

large $d\tau/dt$ amplitudes are determined by large velocities of the promoter, consistently with the large ca. 2.3 °/fs value of the average $-0.5d\delta_{op}/dt$ amplitude seen in Fig. 3a. A large $-0.5d\delta_{op}/dt$ amplitude is not seen in MeO-NAIP@MeOH. In this case, due to the large DTL value and multiple oscillations during such time, it is impossible to enhance the $d\tau/dt^{decay} < 0$ fraction via the synchronization of its 40-fs promoter (a wag corresponding to a limited deformation of the pyrroline and

**Fig. 3 | Distribution of the initial dτ/dt amplitudes. a** Top, progression of α for rPSB11@Rh (left) and MeO-NAIP@MeOH (right). The black lines represent the population motion along $S_1$ before the decay point (represented by red and blue circles). The yellow curves follow the dα/dt average progression, which is further explained using a simple two-component model, where $α_I$ (horizontal dashed line) is associated with a linear progression towards the intersection space and $α_{II}$ associated with the harmonic component of α skeletal twist (or wag). Further details about the α-model can be found in Section V of Supplementary Information. Bottom, same as the top, but for $-0.5dδ_{op}/dt$ progression, a convenient function describing the substituent wag velocity. **b** Comparison of the computed distribution of the dτ/dt (p-orbitals overlap velocity) values of the natural and synthetic switch at 1 fs (left) and 15 fs (right) after population release on the $S_1$ of potential energy surface (PES). The corresponding motion is illustrated by a Newman-like representation in the right plot. The full and open arrows in the left plot represent the velocity variation after 15 fs. The shaded area highlights the dramatic increase of dτ/dt < 0 amplitude. The values in square brackets refer to the isolated chromophores. **c** Same as the previous panel but for the dα/dt (skeletal twist velocity) component. The small dα/dt contribution to the large dτ/dt change is assigned to a $dα_{II}/dt > 0$ value consistently with coupled pyramidalization at C2 and C3. **d** Same as previous panels but for the $-0.5dδ_{op}/dt$ oscillatory component.

indanylidene five-membered ring and therefore analog to HOOP). The $Φ^{iso}$ is then further reduced by the fact that the velocity threshold condition above is largely not satisfied. Below, we show that, in the isolated (i.e. gas-phase) MeO-NAIP, the selection of a different promoter with a longer ca. 250 fs period leads to a $Φ^{iso}$ enhancement.

## Impact of the (catalytic) molecular environment

The effects of the molecular environment on rPSB11 and MeO-NAIP have been studied by computing their dynamics in isolated conditions (see Supplementary Data 2 and 4 for the corresponding numerical data). To ensure an informative analysis, the negative (CCW) pre-twisted conformations found in rPSB11@Rh and MeO-NAIP@MeOH are used as starting points for the generation of the trajectory initial conditions (see also Section VI of the Supplementary Information).

The comparison of Figs. 4a and 2a show that the isolated rPSB11 decays via C11=C12 bond rotation with IDT and DTL close to the ones observed in rPSB11@Rh. This points to a reaction force field intrinsic to the chromophore. Similar to rPSB11, the population decay of MeO-NAIP resembles the one of MeO-NAIP@MeOH but with IDT and DTL ca. 50 fs shorter and 100 fs longer, respectively (compare Figs. 4b and 2b). Further analysis shows that a small rPSB11 trajectory subset, C11=C12 and C9=C10, rotate simultaneously and in opposite directions consistently with Warshel's bicycle-pedal mechanism[25]. Even fewer trajectories lead to C9=C10 isomerization. We conclude that, in rPSB11@Rh the protein enforces a selective C11=C12 isomerization in the CCW direction. Notice that in MeO-NAIP@MeOH such selectivity is imposed by the ring-locked carbon skeleton and CCW pre-twisting. Therefore, both reaction timescale and selectivity in MeO-NAIP are intrinsic properties. The same results support the hypothesis that $S_1$ nuclear coherence still operates in the isolated chromophores. In fact, as shown in the bottom panel of Fig. 4a, PSB11 displays a ca. 50 fs periodic production of reactive trajectories. Most importantly, as we now discuss, a putative $S_1$ mode associated with the ca. 250 fs period of product formation displayed in Fig. 4b is detected in the isolated MeO-NAIP.

The $Φ^{iso}$ values for the C11=C12 isomerization of rPSB11 (we consider the C9=C10–C11=C12 isomerizing trajectories as part of the same reactive population, increasing the computed $Φ^{iso}$ by 9%) and MeO-NAIP are reported in Table 1. Comparison with the corresponding rPSB11@Rh and MeO-NAIP@MeOH values show that the environment induces opposite changes. In Rh, the protein cavity causes a ca. 15% increase (a catalytic effect), while in MeO-NAIP@MeOH the solvent causes a ca. 30% decrease (an inhibitory effect). Table 1 also shows that in both isolated rPSB11 and MeO-NAIP, the reactive trajectories are more loosely associated with the dτ/dt < 0 necessary condition (a ca. <10% decreased confidence) with respect to "solvated" chromophores. This is also seen in Fig. 4c, which displays a diminished relationship between dτ/dt <0 values and reactivity when compared with Fig. 2c. This observation is rationalized considering the difference in the electronic structure of the isolated and cavity-embedded chromophores that we attribute to a counterion effect. In fact, in both rPSB11@Rh and MeO-NAIP@MeOH, the positive charge in the Schiff base moiety is stabilized by a negative counterion. This can be a carboxylate group placed near the C=N moiety as in the Rh cavity or a

virtual counterion formed by oriented methanol molecules (i.e. oriented dipoles) near the C=N moiety of the solvated MeO-NAIP pyridine ring. As illustrated in Fig. 4d for the synthetic rotor, when such counterion is absent, the positive charge gets more delocalized, also inducing a change in the BLA values of the Schiff base fragments. Due to the increased delocalization, the broken (i.e. reacting) π-bond does not reconstitute immediately after the decay, giving time to a part of the reacting molecules to change the dτ/dt phase. This can be seen in Fig. 4e for MeO-NAIP, showing that 15 fs after the decay, the relationship between reactivity and the dτ/dt < 0 condition is reinstated.

It is now necessary to provide a mechanistic interpretation of the opposite $Φ^{iso}$ changes seen when passing from the isolated to the solvated rotors. We start by looking at the protein cavity effect in rPSB11@Rh where the π-bond delocalization impacts the promoter velocity. Such electronic effect is supported by the data in Fig. 5a contrasting $-0.5dδ_{op}/dt$ amplitude in the solvated and isolated environments and showing a substantially decreased amplitude in rPSB11 that must lead to a decreased $dτ/dt^{decay} < 0$ amplitude with respect to rPSB11@Rh. The amplitude decrease appears to be the main cause of $Φ^{iso}$ decrease. Such an effect is also documented by the −1.9 to −0.5 decrease in $-0.5dδ_{op}/dt$ seen after 15 fs of $S_1$ relaxation in the isolated rPSB11 (see Fig. 3d).

MeO-NAIP displays a relatively large 30% increase in $Φ^{iso}$ with respect to MeO-NAIP@MeOH. However, the comparison in Fig. 5b only reveals a slight increase in coherence and amplitude in the critical ca. 40–60 fs oscillatory phase of $dδ_{op}/dt$. On the other hand, a substantial difference in $Φ^{iso}$ must be the consequence of an increase in the fraction of the trajectories featuring $dτ/dt^{decay} < 0$ consistently with the data in Table 2. Furthermore, while due to the small number of reactive trajectories, a periodic product formation in MeO-NAIP@MeOH cannot be detected; the isolated MeO-NAIP displays a single "wave" of reactive trajectories (see red circles in Fig. 5b) starting around 150 fs and a more modest and partially superimposed wave of unreactive trajectories starting around 100 fs after photoexcitation (see blue circles in Fig. 5b). The time length of the wave suggests that the MeO-NAIP reactivity and, ultimately, $Φ^{iso}$ are controlled by a promoter with a longer period with respect to the 40–50 fs mode operating in rPSB11. We now discuss the results that indicate that such a promoter corresponds to a ring-inversion (i.e. an out-of-plane motion or puckering or inversion) coordinate of the pyrrolinium ring and that the lack of such motion decreases $Φ^{iso}$.

Figure 5c displays a schematic representation of the ring-inversion mode potentially impacting $dτ/dt^{decay}$. This is described by the pyrroline N–C2–C3–C4 dihedral (ρ), possibly coupled with the indanylidene C1'–C2'–C3'–C4' (γ) dihedral. Remarkably, the ρ velocity in Fig. 5d shows that its phase correlates with the MeO-NAIP reactivity. In fact, a decreasing dρ/dt appears to correlate with unreactive decays (blue circles) starting at ca. 125 fs and persisting up to 250 fs, while the reactive decays (red circles) are instead seen to appear at ca. 175 fs in connection with an increase in dρ/dt and persist up to ca. 300 fs. Notice that reactive and unreactive trajectories are seen to overlap in the 175–250 fs time range. A complete ρ velocity oscillation is only seen in the isolated MeO-NAIP but not in NAIP@MeOH, indicating a stiffer pyrrolinium ring in the solvated rotor. As we now discuss, such

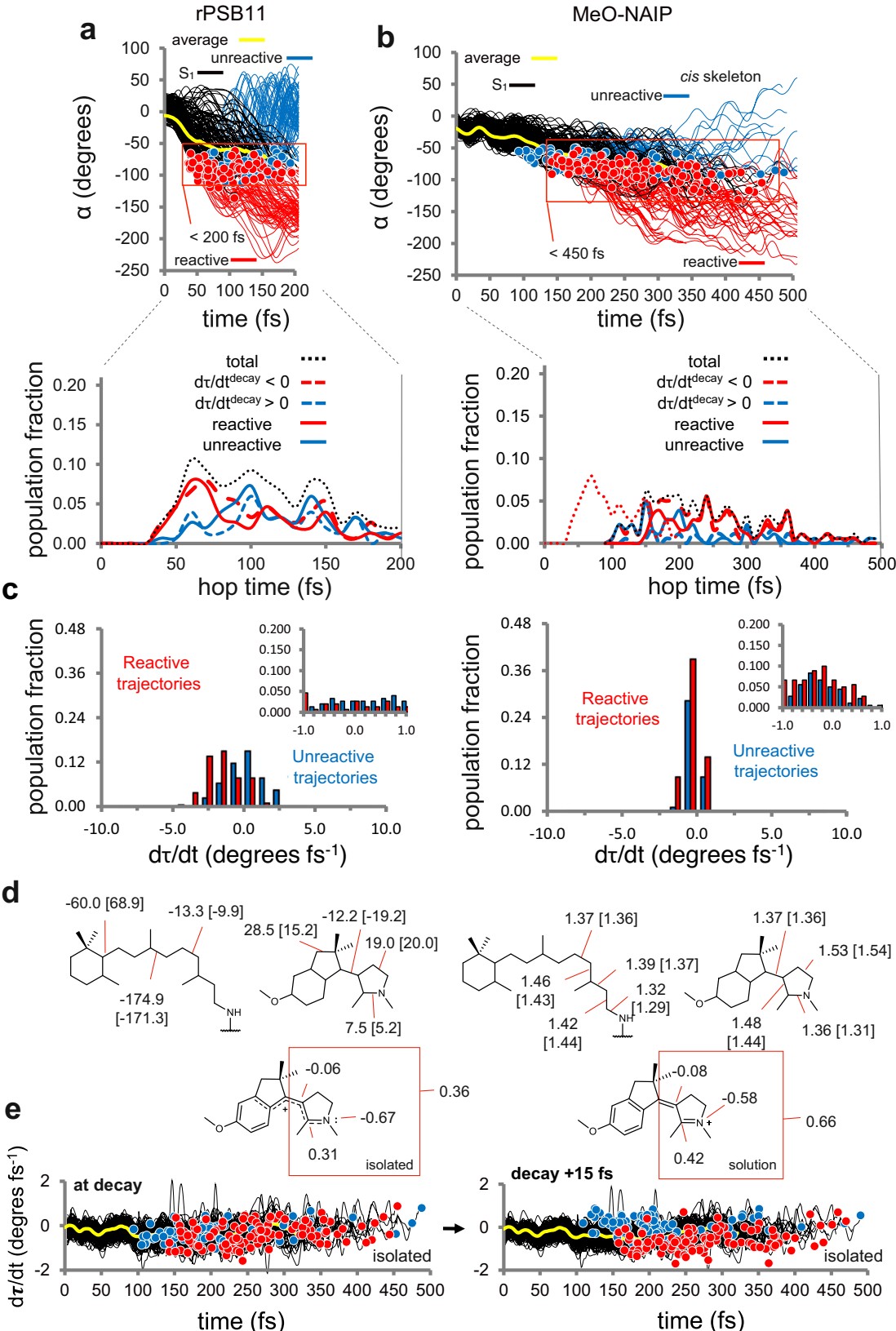

stiffness appears to be induced by a change in the rotor electronic structure, ultimately causing the 30% reduction in $\Phi^{iso}$.

The plot in Fig. 5c shows that the average $d\rho/dt$ is coupled with the average $d\beta/dt$ and, ultimately, $d\tau/dt$ and that this can be a consequence of the ring-inversion (see the Newman projection above the plot. The relationship between $\rho$ and $\beta$ motions is also visualized in the

Supplementary Movies 1–3). Thus, the "mechanism" controlling $\Phi^{iso}$ in MeO-NAIP can be described using the schematic representation of Fig. 6a derived by the population dynamics model mentioned above with the oscillatory component of $\tau$ represented by the sum of two oscillations i and ii. The first has a ca. 40–50 fs period similar to HOOP of Rh, and it is not catalytically effective since, due to the large DTL

**Fig. 4 | Comparison of the photoisomerization dynamics of the isolated rPSB11 and MeO-NAIP chromophores. a** Top: α evolution of rPSB11. The black lines represent the population motion along $S_1$ before the decay point (represented by red and blue circles), and the yellow curves follow the average value. The red lines indicate the trajectories reaching the photoproduct on $S_0$ (reactive trajectories), and the blue lines are the trajectories reaching the original reactant (unreactive trajectories). Bottom: Oscillatory behavior in the reactive trans-product formation. **b** Same data for MeO-NAIP. Less clear oscillations are seen in this case. **c** τ velocity fraction at the decay time (hop time) for the entire population. Reactive trajectories are in red bars, and unreactive trajectories are in blue bars. Most of the reactive trajectories (ca. 80% in both cases) have $d\tau/dt^{decay} < 0$, while a minor number (ca. 20% in both cases) of trajectories with low positive amplitude $d\tau/dt^{decay} > 0$ are

found to be reactive. The bins are 1.0 wide starting from −10.0. The insets show a finer distribution of the bins around the 0.0 value. **d** Top. Most relevant $S_0$ average dihedral angles (left, values in degrees) and bond lengths (right, values in Å) of the two chromophores. The values in square brackets refer to the isolated chromophores. Bottom. Relevant MeO-NAIP (left) and MeO-NAIP@MeOH (right) average $S_1$ charges (in e units) on the pyrrolinium moiety. The total charge of the framed moiety is also given. **e** Time progression of the τ velocity for MeO-NAIP showing the distribution of reactive (red circles) and unreactive (blue circles) trajectories at the hope point (right) and 15 fs after the hop. The black lines represent the population motion along $S_1$ before the decay point (represented by red and blue circles). The yellow curves follow the average value.

value, it is associated with the mechanism of Fig. 6b yielding similar $d\tau/dt^{decay} < 0$ and $d\tau/dt^{decay} > 0$ fractions. Thus, such a high-frequency mode is not the promoter in MeO-NAIP. The second component has a ca. 250 fs period, and it is catalytically effective, leading to an increased $d\tau/dt^{decay} < 0$ population fraction by, presumably, synchronizing its phase with that of dα/dt. This second component would be associated with $d\rho/dt$, which acts as the promoter in MeO-NAIP (see Section VI of the Supplementary Information). Consequently, and consistently with the data in Fig. 5d top, when $d\rho/dt^{decay}$ starts to change towards positive values, a wave of reactive trajectories is generated. The specific change in reactivity of the decay points has also been visualized as a function of the value of $\rho$ (rather than $d\rho/dt$) in Fig. 5e, where the reactive trajectories are seen to populate the increasing part of the $\rho$ curve. Notice that the ring-inversion of the indanylidene five-member ring (angle γ) could contribute to modulating $\Phi^{iso}$, but its motion, although coupled with that of the pyrroline ring, has a longer period and only half of an oscillation appears to be completed during the $S_1$ lifetime (see Fig. 5f). A coupled five-member ring indanylidene and pyrroline ring-inversion motion in MeO-NAIP@MeOH has been reported in earlier publications[3], but the corresponding $S_1$ population isomerization dynamics has never been reported before.

The putative "catalytic" ring-inversion motion found in MeO-NAIP is quenched by the solvent. Such quenching appears to be related to the presence of a virtual counterion near the N atom of the pyrroline moiety, leading to the generation of a stronger C3=C4 double bond in the pyrroline ring and a higher ring stiffness accompanied by a significant ring-inversion barrier blocking the ring-inversion. This has been investigated by (i) comparing the initial charge distribution (see Fig. 4d) and its time evolution (see Fig. 5g) for MeO-NAIP and MeO-NAIP@MeOH and (ii) by constructing two $S_0$ model systems (see Fig. 5h) mimicking the different electronic structure of the pyrroline moiety. The first model, a neutral enamine, mimics a pyrroline moiety featuring a $sp^3$ N atom and a localized double bond in position C3=C4 hypothesized to be the dominant electronic structure of a solvated MeO-NAIP in the $S_1$ state. Consistently with the plot in Fig. 5g bottom, such an electronic structure is generated via the large charge-transfer motion occurring upon $S_1$ relaxation in the solvated rotor. The second, a cyanine, mimics a partially positively charged pyrroline moiety featuring highly delocalized N=C2 and C3=C4 double bonds consistently with the charge plot in Fig. 5g top for an isolated MeO-NAIP in the $S_1$ state. As detailed in Section VI of the Supplementary Information, we find that the vibrational excitation of the first model does not lead to a ring-inversion or puckering motion within 500 fs, while at least one full inversion is seen in the second model, like that seen in the MeO-NAIP plot. It is therefore proposed that the $\Phi^{iso}$ inhibitory effect is caused by the virtual counterion generated by the solvent, and it is essentially due to the change in the electronic structure of the $S_1$ chromophore quenching the ring-inversion motion. Such an inhibitory effect has not previously been described and is markedly different from the mixing of the $S_1$ and $S_2$ states previously reported[26].

The contribution of a steric, rather than electronic effect to the quenching of the ring-inversion motion MeO-NAIP@MeOH cannot be

excluded. In order to partially investigate this process, we have run, using exactly the same modeling methods, MeO-NAIP population dynamics (see Supplementary Data 5 for the corresponding numerical data) in the solvent dimethyl sulfoxide (DMSO). DMSO is a polar aprotic solvent with higher viscosity (ca. four times) with respect to MeOH. Comparison between these solvent environments would thus allow us to test (a) the effect of a higher viscosity as well as (b) the lack of hydrogen bonding. As detailed in Section VI of the Supplementary Information, while MeO-NAIP@DMSO shows a qualitative behavior like the one in MeOH, the percentage of the population undergoing ring-inversion in the pyrrolinium moiety is slightly increased, most probably due to a change in $S_1$ electronic structure that may be hypothetically attributed to a decreased stabilization of the positive charge due to lack of hydrogen bonding. The increase in viscosity (i.e. a steric factor, presumably leading to a decreased ring-inversion motion) does not seem to be implicated in the described behavior.

The classification of molecular-level factors determining the quantum efficiency of elementary photochemical reactions is highly needed not only because of their fundamental importance but also for the obvious applications in molecular engineering. Above, we have reported on a research effort investigating such factors for the case of homolog ultrafast double bond isomerization in a biological photoreceptor and a synthetic molecular rotor sharing the same organic function. The comparative analysis of hundreds of quantum-classical trajectories revealed that not only the phase but also the amplitude of $d\tau/dt^{decay}$ has an impact on $\Phi^{iso}$. Indeed, we have provided evidence for the existence of a "statistically" sufficient condition for reactive trajectory, indicating that high $\Phi^{iso}$ can only be achieved by maximizing both the $d\tau/dt^{decay} < 0$ population fraction and its absolute amplitude within the coherent (high-speed) phase-driven mechanism of Fig. 6c. According to our calculation, this is how Nature has selected rPSB11@Rh, while the synthetic rotor MeO-NAIP@MeOH has a 50% $d\tau/dt^{decay} < 0$ population fraction and a low (<1.0 degree/fs) absolute amplitude yielding a lower computed 30% $\Phi^{iso}$ pointing to the mechanism of Fig. 6b.

The geometrical and electronic determinants of the established sufficient condition for reactivity have been investigated by developing a basic model connecting the geometrical factors α and $\delta_{op}$ to $d\tau/dt$. We show that a large $d\tau/dt^{decay} < 0$ fractions and amplitude requires a periodic mode called promoter synchronized with the time needed to access the decay region and capable of generating high amplitude $-0.5d\delta_{op}/dt^{decay} < 0$ power-stroke motions. These requirements cannot be achieved in a motor like MeO-NAIP@MeOH where, simply, a suitable promoter mode does not exist, and the homolog HOOP promoter spans several $-0.5d\delta_{op}/dt$ oscillations.

The mechanistic principles discussed above have been used to rationalize the impact of molecular cavities on the $\Phi^{iso}$ value computed for the isolated rPSB11 and NAIP. The catalytic effect of the Rh cavity is ascribed to its role in increasing the selectivity of the reaction, enhancing the reaction vibrationally coherence, and increasing the $d\tau/dt^{decay} < 0$ population fraction via the HOOP wag promoter. In contrast, the inhibitory effect induced by the substantially unstructured MeOH

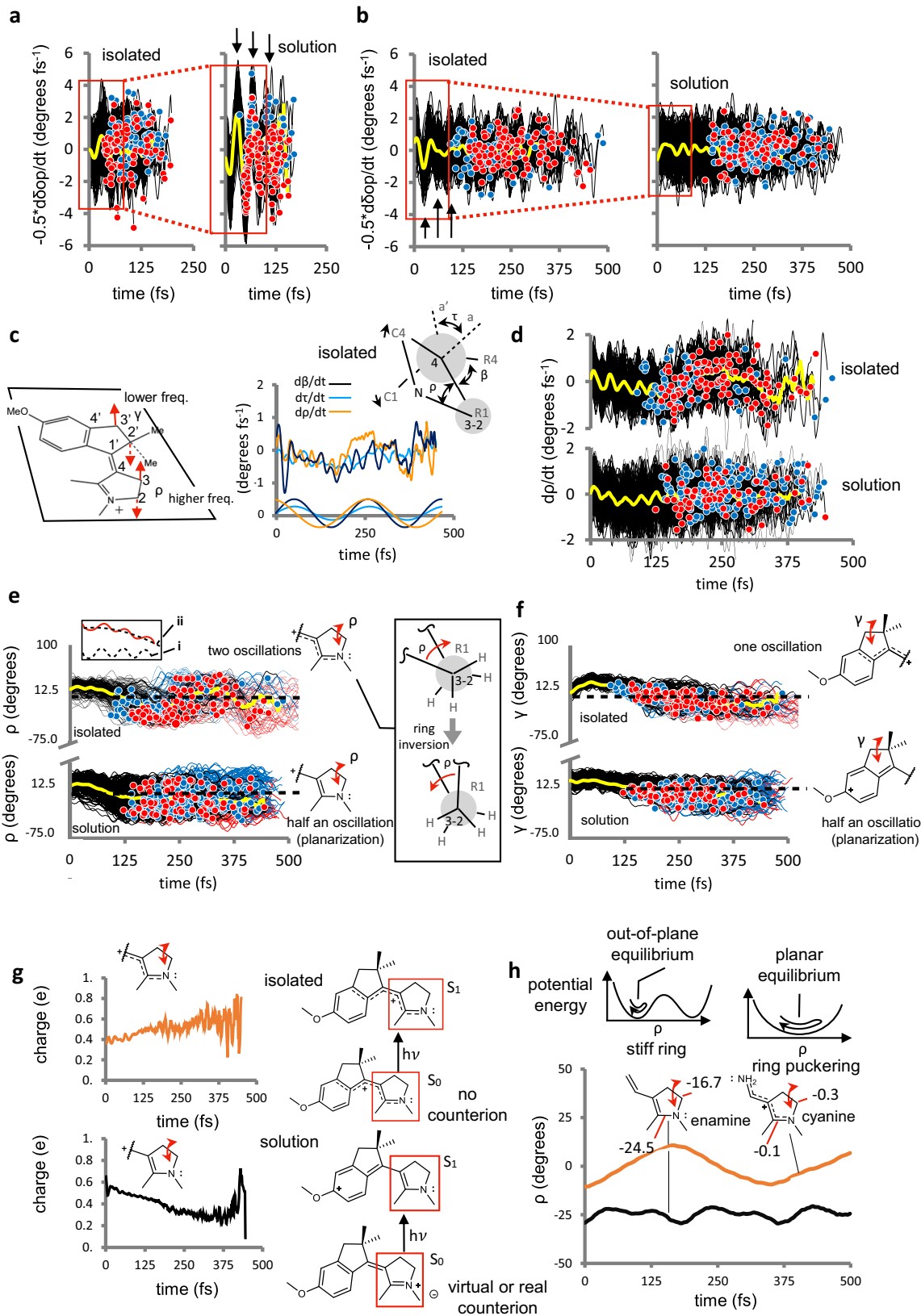

cavity is attributed to the quenching of the potential ring-inversion promoter through an electronic effect induced by a counterion. Such $\Phi^{iso}$ effect is totally absent in rPSB11, does not have five-member rings incorporating R1, R2, and C1, C2, and whose counterion electrostatic effect is partially counterbalanced by the rest of the protein residues[27].

In the past, the impact of the HOOP wag, a mode not related to the reaction coordinate, on the isomerization quantum efficiency of the rPSB11@Rh rotor has been seen as a property of a specific class of molecules (i.e. the rhodopsin protein family). However, above, we have shown that not only a promoter mode, is found in a synthetic rotor, but that: (i) it is a qualitatively different mode featuring an order of

**Fig. 5 | Quantum efficiency changes in isolated rotors. a** Promoter velocity changes in isolated and "solvated" rPSB11. **b** Same changes in MeO-NAIP. The yellow curves follow the average progression; the same applies to the plots in panels **a**, **b**, **d**, **e**, and **f**. **c** Left: schematic representation of the ring-inversion of the pyrrolinium ($\rho$) and indanylidene ($\gamma$) moieties. Right: time progression of the average $\rho$ velocity in isolated conditions and correlation with the average $\tau$ and $\beta$ velocities. The sinusoidal fitting of the three average velocities is reported at the bottom and displays periods of 261, 200, and 204 fs for $\rho$, $\beta$, and $\tau$, respectively. The Newman projections at the top provide a schematic representation of the coupling between $\rho$, $\beta$ (thus, $\delta_{op}$), and $\tau$ (see also Section VII in the Supplementary Information). **d** Time progression of d$\rho$/d$t$ (i.e., pyrroline moiety ring-puckering of MeO-NAIP (top) and MeO-NAIP@MeOH (bottom). **e** Progression of the corresponding $\rho$ angle. The resonance formulas on the right inform on the dominant electronic structure imposed by the environment. **f** Same progressions for $\gamma$. **g** Time progression of the charges on the pyrrolinium moiety for both MeO-NAIP (top) and MeO-NAIP@-MeOH (bottom). **h** Time progression of the enamine (left) and cyanine (right) models representing the limiting electronic structures of the MeO-NAIP pyrrolinium unit.

magnitude lower frequency. In fact, the analog HOOP mode does not operate in the synthetic system as it cannot synchronize with the slower α progression. To synchronize with such a timescale, a ring-inversion mode (ρ) is selected in the isolated MeO-NAIP. (ii) The existence of a promoter mode of the right frequency is determined by the rotor $S_1$ electronic structure. Thus, factors affecting the electronic structure (e.g. substituents or solvents) can modulate $\Phi^{iso}$ through such a mechanism.

Collectively, the above findings reshape our understanding of how quantum efficiency works at a fundamental level by demonstrating that $\Phi^{iso}$ depends, again, not only on rotary motion but also on a synchronized auxiliary molecular promoter mode. This provides a rational explanation for the observed diminished $\Phi^{iso}$ of a synthetic light-driven molecular rotor in methanol compared to a natural homolog designed by Nature. Moreover, the information obtained by comparing the dynamics of the two systems clarifies the relationship between the d$\tau$/d$t^{decay} < 0$ reactivity condition and the Landau–Zener model, thus connecting our work with the foundation of photochemistry.

The theoretical framework presented in this work, based on an extensive quantum-classical trajectory study using QM/MM modeling, has been able to relate relatively simple geometrical and electronic factors controlling the $\Phi^{iso}$ value of a chemical reaction of basic importance in photobiology and for various molecular devices. We are convinced that this will soon lead to the computational design and preparation of novel chemically modified NAIP-like biomimetic motors showing a significantly increased $\Phi^{iso}$.

## Methods
### QM/MM and QM modeling and validation
Our research is based on quantum-mechanics/molecular-mechanics (QM/MM) models of rPSB11[13,14] embedded in the cavity of Rh (rPSB11@Rh) and the synthetic MeO-NAIP[1–3] in methanol solution (MeO-NAIP@MeOH). As evident from the structures reported in Fig. 1a, the rPSB11 and MeO-NAIP conjugated Schiff base cations differ in (i) the conjugated chain (e.g. a long linear chain rather than a short chain terminated by a phenyl with a *para* electron releasing group), (ii) the substitution pattern (e.g. methyl groups vs. alkyl bridges forming strained cycles) and (iii) the molecular environment (a protein cavity rather than a solution environment).

In order to treat the differences i–iii consistently, we adopted the following protocols. The QM/MM model of rPSB11@Rh was constructed starting from the crystallographic structure available in the Protein Data Bank (PDB ID 1U19)[28] following a previously reported protocol[15]. The QM subsystem comprises the retinal chromophore, NH, and C$\varepsilon$H3 atoms linked to the C$\delta$ atom of the Lys296 sidechain, which is treated at the complete active space self-consistent field (CASSCF) level of theory with the 6–31G* basis set (CASSF/6–31G*). The selected active space of 12 electrons in 12 orbitals comprising the full π system of rPSB11. The rest of the protein defines the MM subsystem, which is described by a modified AMBER94 force field featuring specific parameters for the Lys296 side-chain[29,30]. All side chains or waters within 4 Å from the chromophore atoms were free to relax during the calculation, while the MM atoms were kept frozen. The model of MeO-NAIP@MeOH was

constructed via classic molecular dynamics equilibration with periodic boundary conditions. The entire MeO-NAIP molecule defines the QM subsystem, which is treated at the same CASSCF/6–31G* level used for Rh and with 12 electrons in 11 orbitals active space corresponding to the entire π-system of MeO-NAIP. The MM subsystem is formed by the solvent molecules described by OPLS-aa force field parameters[31]. Consistently with the Rh model, the solvent molecules within 4 Å from the QM subsystem are kept flexible while the rest of the solvent atoms are frozen during the simulation.

The isolated rPSB11 and NAIP chromophores are modeled at the QM level using the treatment applied to the corresponding QM subsystems described above. In the case of rPSB11@Rh and rPSB11, the introduction of the $S_2$ in the CASSCF (i.e. $S_0$, $S_1$ and $S_2$ state averaging) calculation makes the $S_1$ PES flatter, generating too long excited state lifetimes. On the other hand, the interaction of $S_1$ and $S_2$ has been proven to be limited (see details in Section I of the Supplementary Information). That is why, as the best compromise, we use $S_0$, $S_1$, state averaging. This does not apply to MeO-NAIP where $S_0$, $S_1$, and $S_2$ state averaging is used. The reason is that the synthetic system has higher excitation energies (it absorbs UV light rather than visible light) and features an initially smaller $S_2$-$S_1$ energy gap (many initial conditions for CASSCF trajectory calculations show $S_2$ being the spectroscopic state). Note that in both cases, the adopted strategy produces results consistent with the experimental observations.

### Initial conditions and population dynamics simulations, excited state lifetime
The 200 initial conditions (geometries and velocities) defining the $S_0$ populations at room temperature are generated for both systems via a protocol reported earlier[15]. Briefly, MM dynamics simulation at 298 K was initiated from an $S_0$ optimized geometry reproducing the wavelength of the corresponding absorption maxima ($\lambda_{max}$). 200 snapshots (geometries and velocities) were then extracted during the dynamics at fixed time intervals after a suitable equilibration time. Starting from these snapshots, 200 HF/631G*/Amber and CASSCF/6-31G*/OPLS-aa levels for Rh and MeO-NAIP, respectively, were propagated for 200 fs and then followed by corresponding $S_0$ 2-root-sate-average CASSCF/6-31G*/Amber trajectories for 50 fs for the case of Rh. The 200 geometries and velocities of the final snapshot of the 50 and 200 fs CASSCF propagation, respectively, are assumed to represent, for each system, the Boltzmann distribution (to best account for the PES anharmonicity, a Boltzmann or Wigner sampling based-on the Hessian Matrix were not considered) and, therefore, the initial conditions for subsequent quantum-classical trajectory computations starting on $S_1$. The quantum-classical trajectories were propagated at the 2-root-state-average CASSCF/6–31G*/Amber and 3-root-state-average CASSCF/6–31G*/OPLS-aa level (for Rh and MeO-NAIP, respectively) following the Tully surface-hopping method including the decoherence correction[18,19]. (A test supporting the validity of the Tully surface-hopping method for alkylated or protonated Schiff bases has been recently reported[32]). All calculations were performed using Molcas/Tinker package[33,34]. In order to determine the $S_1$ lifetime reported in Table 4, we have used the following fitting formula comprising a lag time and two exponential decay times. Further details about the initial condition generation and trajectory calculation are documented in

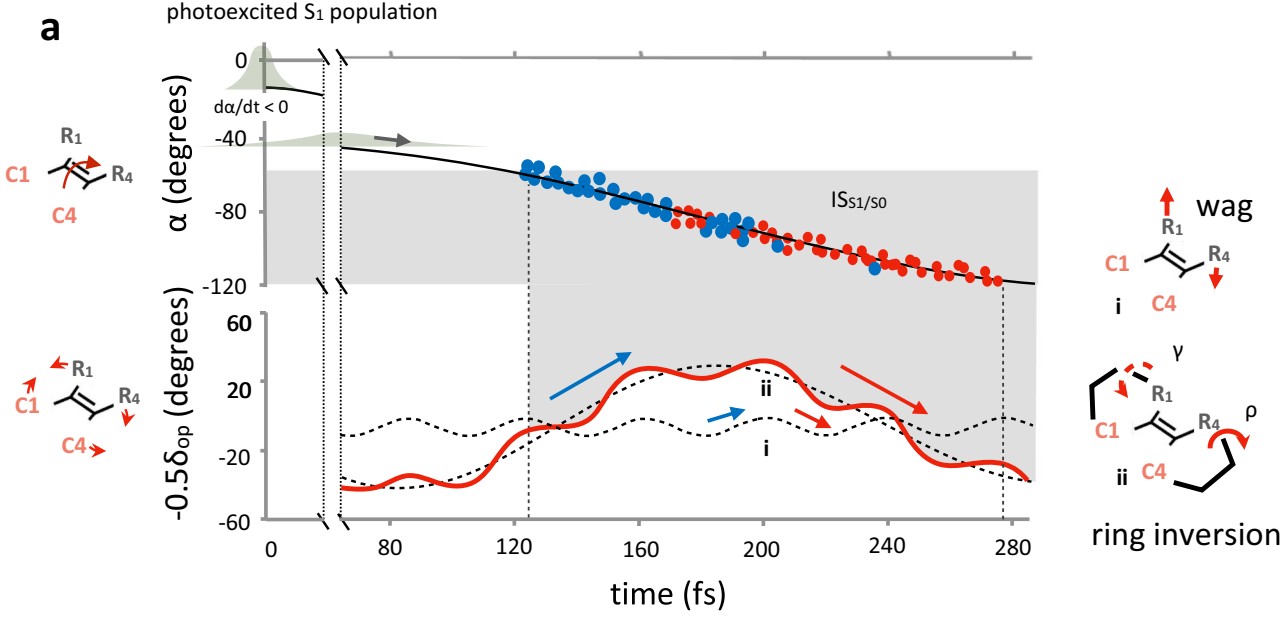

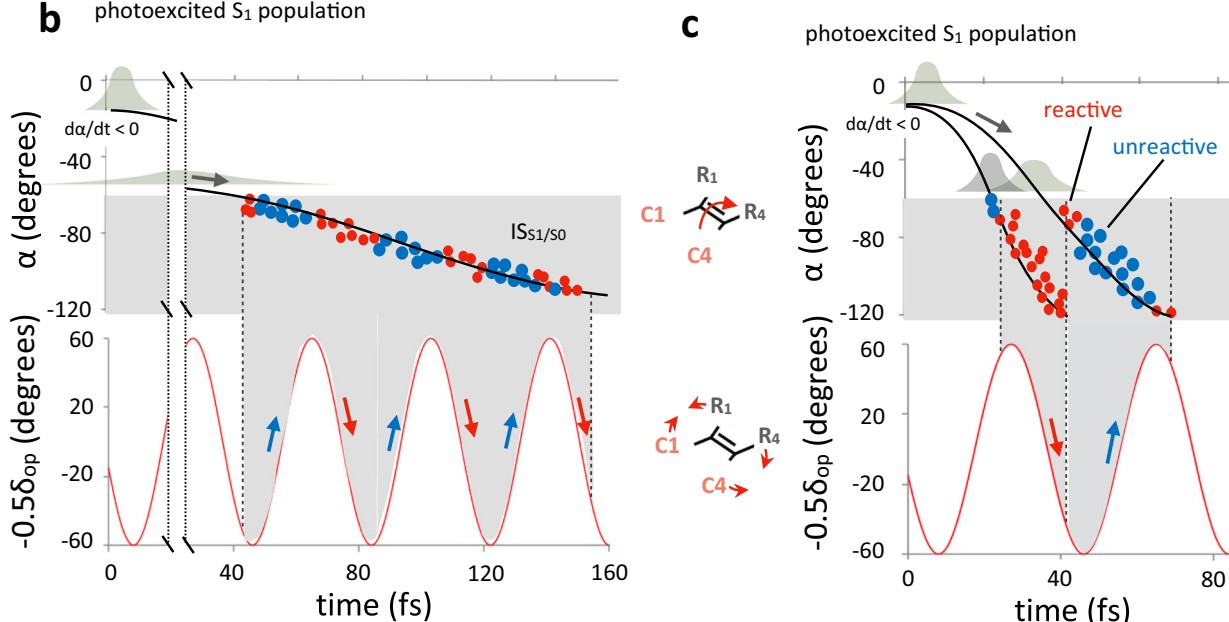

**Fig. 6 | Slow and fast model progression of the monotonic ($\alpha$) and oscillatory ($-0.5\delta_{op}$) components of $\tau$. a** *Top*, Schematic representation of $\alpha$ progression for a population decaying slowly (represented by broad Gaussian profiles). The dark gray area corresponds to the intersection space or decay region ($IS_{S1/S0}$). The red circles represent reactive decay events, while the blue circles represent unreactive events. *Bottom*, Corresponding time progression of $-0.5\delta_{op}$ resulting from the superposition of one low-amplitude high-frequency (similar to the Rh hydrogen out-of-phase, HOOP mode) and one high-amplitude low-frequency (e.g., resulting from an $S_1$ ring-inversion motion, $\rho$ mode labeled i and ii (see dashed curves) respectively. The wide distribution of $\alpha$ velocities in the $IS_{S1/S0}$ region leads to a decay ultimately regulated by the phase (blue and red arrows) of the low-frequency mode, leading to a single "wave" of trajectories with $d\tau/dt^{decay} < 0$. **b** Same as the previous panel, but for a population characterized by a single high-frequency

$-0.5\delta_{op}$ oscillatory component. In this case, the wide distribution of $\alpha$ velocities leads to a population decaying with alternating phases, leading to multiple population fractions decaying with $d\tau/dt^{decay} < 0$ and $d\tau/dt^{decay} > 0$. The presence of a high-frequency component leads to several trajectory fractions with alternative negative and positive phases (red and blue arrows) and similar $d\tau/dt^{decay} < 0$ and $d\tau/dt^{decay} < 0$ fractions. **c** Same as for the previous two panels but for two different fast-reacting populations (represented by Gaussian profiles) and one a high-frequency $-0.5\delta_{op}$ oscillatory component. The diagram shows that two slightly different initial conditions, both featuring a narrow $\alpha$ distribution, may lead to a single reactive or a single unreactive trajectory fraction. The negative phase (red arrow) corresponding to the fastest population leads to a large (left) $d\tau/dt^{decay} < 0$ fractions.

Section I of the Supplementary Information[35].

$$f(t) = a_1 e^{\left(-\left(\frac{t-t_1}{t_2}\right)^2\right)} + y_0 + ((1 - a_1) - y_0)e^{\left(\frac{-(t-t_1)}{t_2}\right)} \tag{1}$$

To study the protein and solve induced environment effects, we repeat the above initial condition calculations for the two isolated chromophores (i.e., in the absence of the protein and methanol solvent

**Table 4 | Comparison between measured and computed spectroscopic and dynamics properties**

| rPSB11@Rh | $\lambda_{max}$ (nm) | Product appearance time (fs) | Lifetime (fs) | Vibrational freq. (cm$^{-1}$), $\delta_{op}$ | vibrational freq. (cm$^{-1}$), BLA | $\Phi$cis–trans (%) |
|---|---|---|---|---|---|---|
| Obs. | 498 | <50[a] | ~80[b] | 746[a] | 1679[a] | 67[c] |
| Comp. | 500 | > 70[d-i] | 98[d-ii] (93[d-iii]) | 855, 1068[d] | 1588[d] | 68[d] 72[e] (69[f]) |
| Comp. Isolated | 560 | – | 114[d-ii] (107[d-iii]) | – | – | 47[d] |
| **MeO-NAIP@MeOH** | $\lambda_{max}$ (nm) | Product appearance time (fs) | Lifetime (fs) | Vibrational freq. (cm$^{-1}$), $\delta_{op}$ | Vibrational freq. (cm$^{-1}$), BLA | $\Phi$cis–trans (%) |
| Obs. | 389[g] | >300[h] | 260[h] | Not-determined | 1572[i] | 21[j] |
| Comp. | 383 | 358 | 281[d-ii] (265[d-iii]) | 565, 847[d] | 1450[d] | 30[d] |
| Comp. Isolated | 480 | – | 254[d-ii] (240[d-iii]) | – | – | 55[d] |

[a]Observed data from ref. 9.
[b]Observed data from ref. 36.
[c]Observed data from ref. 14.
[d]Based on a 200-trajectory ensemble.
[d-i]Time range from the first successful decay to the first photoproduct with a bathoRh-like/E-isomer absorption.
[d-ii]Fitted S$_1$ lifetime of 200 trajectories.
[d-iii]Average decay time of 200 trajectories.
[e]Based on a 50-trajectory sample with a Rh model of a full rPSB11 chromophore.
[f]Based on a 400-trajectory sample with a Rh model of a reduced rPSB11 chromophore from ref. 15.
[g]Observed data from ref. 1.
[h]Observed data from ref. 12.
[i]Observed data from ref. 3.
[j]Observed data from ref. 2.

for Rh and MeO-NAIP, respectively). Therefore, the sampling is limited to the isolated retinal chromophore with a terminal –C15=N–C$_6$H$_3$ group and to the MeO-NAIP cation that is ultimately modeled at the CASSCF/6–31G* level of theory. Further details about the QM/MM and QM model constraints and validation are documented in Sections I–III of the Supplementary Information.

### Statistical analysis

An important part of the present study is the statistical analysis of the rPSB11@Rh and MeO-NAIP@MeOH hop points. Accordingly, the following properties have been determined at each hop point:

A. reactive vs. unreactive hop.
B. negative vs. positive $d\tau/dt$,
C. magnitude of $d\tau/dt$

All velocities are calculated numerically in terms of the differences between the values of successive 1 fs time steps. We use properties A–C to classify the molecular population in different subsets. Of course, we are mostly interested in the properties of the reactive subpopulations, which are defined as the group of trajectories achieving the product geometry after the hop. These subpopulations are then analyzed based on the corresponding $d\tau/dt$ sign and amplitude. Further details can be found in Section IV of the Supplementary Information.

### Reporting summary

Further information on research design is available in the Nature Portfolio Reporting Summary linked to this article.

## Data availability

The authors declare that the data supporting the findings of this study are available within the main article and the Supplementary Information. Cartesian coordinates generated along the trajectories can be found at https://zenodo.org/records/5826280. Source data are provided in this paper. Source data are provided with this paper.

## Code availability

The authors declare that the present research has been produced with distributed software available to the public, as also detailed in the Supplementary Information.

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

## Acknowledgements

M.O. is grateful to the NSF CHE-SDM A for Grant No. 2102619. This research was, in part, supported by EU funding within the MUR PNRR "National Center for Gene Therapy and Drugs based on RNA Technology" (Project no. CN00000041 CN3 RNA)—Spoke 6. M.O. is also grateful to BGSU and the Center for Photochemical Sciences for additional support and the Ohio Supercomputer Center for the computational infrastructure.

## Author contributions

A.B.-G., M.M., and X.Y. created the models and performed the simulations. A.B.-G., X.Y., and M.O. analyzed the results. M.O. and A.B.-G. have written the manuscript. M.O. supervised the study, and all authors contributed to the manuscript.

## Competing interests

The authors declare no competing interests.
