## [Peer Review File · Nature Communications]

REVIEWER COMMENTS

Reviewer #1 (Remarks to the Author):

I thank the authors for addressing my comments. There might be room for further discussions on some of their answers, but the warnings added to the main text would be sufficient for the reader to be informed on some open questions. I would recommend this manuscript for publication in Nature Communications.

Reviewer #2 (Remarks to the Author):

Dear Editor,

I acknowledge the significant revision work performed by the authors of the article "Comparative Quantum-Classical Dynamics of Natural and Synthetic Molecular Rotors Shows How Vibrational Synchronization Modulates Quantum Efficiency" and kindly inform you that part of my concerns have been now addressed. However, focusing on the most important motivations behind this work which should support its impact in the community and justify publication on Nature Comm, there are still few pending aspects that make me doubtful about its acceptance.

Specifically, the authors have been stressed in the revised text that this article is of interest to a large part of the photochemistry scientific community, because it is focusing on (and analyzing the) factors controlling the quantum efficiency of photochemical reactions. If properly addressed, this would certainly be a good motivation for the work to be published in Nature communications, but at the same time I am still not convinced that the model with which they explain the efficiency of the new light-driven biomimetic molecular rotor MeO-NAIP is well rationalized and/or supported by the presented data. So, at the end, this work is not yet delivering a clear and consistent model. The two main reasons why I do not still consider this model to be enough solid and why I did not get convincing answers are explained below.

1. My main doubt is still concerning the defined main reactivity condition ($d\tau/dt$ decay < 0 values), which is reasonably addressed to retinal chromophores, but, in my opinion, not to the MeO-NAIP biomimetic system.

Although I appreciate the corrections and improvements made by the authors on Figure 2c and Figure 4c, corresponding to dynamical results in solvent and in gas phase, including the addition of the insets displaying a finer view of the important range around zero velocities, I'm still not clearly catching the $d\tau/dt < 0$ reactive rule in that region, in the case of the biomimetic system.

In fact, for the retinal we can clearly observe that in the 'reactive region' ($d\tau/dt < 0$) the red bars (relating to reactive structures) are on average clearly higher than the blue bars (non-reactive structures), and the exact opposite occurs in the 'non-reactive region' ($d\tau/dt > 0$).

However, this is not true for the biomimetic system, for which we see that in Figure 2c the 'reactive region' shows blue and red bars almost equivalent, whereas in the 'non-reactive region', although the blue bar is higher, its height does not reflect the details shown in the corresponding inset: in fact, the red bars seem to prevail in the inset panel, at odd with what shown in the main picture.

Moreover, in the 'non-reactive region' of Figure 4c (gas phase) the reactive bar of the MeO-NAIP is higher than the unreactive one.

This makes me think that the discriminating parameter $d\tau/dt < 0$ (in the region of values between -1 and +1) does not clearly represent the reactive character of the biomimetic system, even when

associated with the puckering mode promoter.

An anomaly in the relationship between reactivity and $d\tau/dt < 0$ is possibly confirmed by the graphs representing 'hop time(fs)' values vs. 'population fraction' in Figures 2 and 4. In these graphs, a clear correspondence between $d\tau/dt < 0$ values (solid red line) and reactive trajts (dashed red line) is observed for the retinal case. The same for $d\tau/dt > 0$ values (solid blue line) and unreactive trajts (dashed blue line). Differently, this correspondence is no more clearly subsisting in the MeO-NAIP. Given the great effort of clarification shown by the authors, I suppose their reasoning may be consistent, but perhaps not clear to my eyes. This means that there are expository deficiencies that I would suggest to fix.

2. The second fundamental argument to which I would have liked a convincing answer concerns the puckering promoter mode, which can strongly enhance the QY of the biomimetic system. The authors claimed that this important mode could be actually prevented in solvent by electronic reasons given by 'counterion effects'. My personal doubt was that a reduction in the puckering mechanism could also be eventually induced by the steric hindrance of the solvent. Verification of this doubt seemed to me of some importance, because if the inhibition had steric reasons, the biomimetic system could not work in either polar or non-polar solvents ... possibly only in gas phase, which, from an application point of view, would be really limiting.

The authors tried to address this point by running one single dynamics in the aprotic solvent dimethyl sulfoxide (DMSO) following the same protocol already described for methanol solvent, starting from one randomly selected geometry of the sampled initial conditions. Then, they compared the evolution of the puckering mode on S1 over time in the three different environments, i.e. gas phase, MeOH and DMSO. Since the trajectory in gas phase and in DMSO showed more similarities than that in MeOH, they valued this result optimistic over the hypothesis that it is a 'counterion effect' that prevents puckering promoter mode. But we clearly know that a single dynamic has no statistical meaning. A 0°K dynamics would have better served this cause. Alternatively, a quick, simple, and scientifically valid test could be to optimize the MeO-NAIP system in two specific structures, namely planar and distorted (puckering), in the three different environments respectively. Their corresponding energies clearly could tell us in which environment the puckering is most favored and then statistically more dynamically populated.

In conclusion, my two main doubts with respect to the proposed model are still pending. However, I believe that, if the model is robust, these nodes could be fixed and clearly explained.

RESPONSE TO REVIEWER COMMENTS

Article reference: NCOMMS-23-32330-T

Comparative Quantum-Classical Dynamics of Natural and Synthetic Molecular Rotors Shows How Vibrational Synchronization Modulates Quantum Efficiency

We thank both reviewers for their critique. The following is a point-by-point response. We sincerely hope that our re-revised manuscript is suitable for publication in Nature Communications.

Contents:

Response to Reviewer #1: Page R2

Response to Reviewer #2: Page R3

Blue = Reviewer Comments, **Black** = Authors answers, **Red** = important points and changes to the manuscript. **Green** = new changes

RESPONSE TO REVIEWER #1 COMMENTS

Initial comment:

I thank the authors for addressing my comments. There might be room for further discussions on some of their answers, but the warnings added to the main text would be sufficient for the reader to be informed on some open questions. I would recommend this manuscript for publication in Nature Communications.

Initial response:

We thank the reviewer for appreciating the significance of our work and for recommending this manuscript for publication.

RESPONSE TO REVIEWER #2 COMMENTS

Initial Comment

I acknowledge the significant revision work performed by the authors of the article "Comparative Quantum-Classical Dynamics of Natural and Synthetic Molecular Rotors Shows How Vibrational Synchronization Modulates Quantum Efficiency" and kindly inform you that part of my concerns have been now addressed. However, focusing on the most important motivations behind this work which should support its impact in the community and justify publication on Nature Comm, there are still few pending aspects that make me doubtful about its acceptance.

Specifically, the authors have been stressed in the revised text that this article is of interest to a large part of the photochemistry scientific community, because it is focusing on (and analyzing the) factors controlling the quantum efficiency of photochemical reactions. If properly addressed, this would certainly be a good motivation for the work to be published in Nature communications, but at the same time *I am still not convinced that the model with which they explain the efficiency of the new light-driven biomimetic molecular rotor MeO-NAIP is well rationalized and/or supported by the presented data.* So, at the end, this work is not yet delivering a clear and consistent model.

The two main reasons why I do not still consider this model to be enough solid and why I did not get convincing answers are explained below.

Response:

We thank the reviewer for acknowledging our manuscript's revision and for the constructive feedback. Indeed, our research aims to improve the understanding of reaction quantum efficiency in general. In the presented case, we show that the synchronization of specific vibrational modes (called promoters) with the reaction coordinate provides a general mechanism for quantum efficiency modulation in molecular rotors. In response to the reviewer comments, we have further explained the validity of the proposed general reactivity condition in the case of the MeO-NAIP biomimetic system (see reply to Comment 1), and further investigated the role of its pyrrolinium ring-puckering mode (see reply to Comment 2) as promoter in different environments. We now hope that the added discussion and, most importantly, novel trajectory calculations now reported in the main manuscript and Supplementary Information, fully address the reviewer concerns.

Change/Remark 1: The following sentence on page 2 of the main text emphasizes the importance of the present contribution.

“...building a theoretical framework linking molecular structure and Φ^{iso} variations. The present work attempts to establish such a link that is, not arguably, of basic relevance for the field of photochemistry...”

Change/Remark 2: The following sentence on page 5 of the main text stresses the main target of the presented research.

“...to uncover the origin of the different Φ^{iso} values observed for rPSB11 and MeO-NAIP in different environments....”

Comment 1:

My main doubt is still concerning the defined main reactivity condition ($d\tau/dt$ decay < 0 values), which is reasonably addressed to retinal chromophores, but, in my opinion, not to the MeO-NAIP biomimetic system.

Although I appreciate the corrections and improvements made by the authors on Figure 2c and Figure 4c, corresponding to dynamical results in solvent and in gas phase, including the addition of the insets displaying a finer view of the important range around zero velocities, I'm still not clearly catching the $d\tau/dt < 0$ reactive rule in that region, in the case of the biomimetic system.

In fact, for the retinal we can clearly observe that in the 'reactive region' ($d\tau/dt < 0$) the red bars (relating to reactive structures) are on average clearly higher than the blue bars (non-reactive structures), and the exact opposite occurs in the 'non-reactive region' ($d\tau/dt > 0$).

However, this is not true for the biomimetic system, for which we see that in Figure 2c the 'reactive region' shows blue and red bars almost equivalent, whereas in the 'non-reactive region', although the blue bar is higher, its height does not reflect the details shown in the corresponding inset: in fact, the red bars seem to prevail in the inset panel, at odd with what shown in the main picture.

Response:

This inconsistency in the inset diagram (a mere mistake in plotting the data) with respect to the main diagram, has now been corrected. See **Change 5** for details.

Moreover, in the 'non-reactive region' of Figure 4c (gas phase) the reactive bar of the MeO-NAIP is higher than the unreactive one.

Response:

The presented data are correct but belong to trajectories with absolute $d\tau/dt^{\text{decay}} < 1.0$ degrees/fs. As explained below the $d\tau/dt^{\text{decay}} < 0$ condition is not valid in this low velocity regime that constitutes a threshold for such reactivity condition (see discussion below).

This makes me think that the discriminating parameter $d\tau/dt < 0$ (in the region of values between -1 and +1) does not clearly represent the reactive character of the biomimetic system, even when associated with the puckering mode promoter.

Response:

This conclusion of the reviewer is correct but do not invalidate the general validity of the proposed reactivity theory when considering the above mentioned threshold (see discussion below).

An anomaly in the relationship between reactivity and $d\tau/dt < 0$ is possibly confirmed by the graphs representing 'hop time(fs)' values vs. 'population fraction' in Figures 2 and 4. In these graphs, a clear correspondence between $d\tau/dt < 0$ values (solid red line) and reactive trajts (dashed red line) is observed for the retinal case. The same for $d\tau/dt > 0$ values (solid blue line) and unreactive trajts (dashed blue line). Differently, this correspondence is no more clearly subsisting in the MeO-NAIP. Given the great effort of clarification shown by the authors, I suppose their reasoning may be consistent, but perhaps not clear to my eyes. This means that there are expository deficiencies that I would suggest to fix.

Response:

We are grateful to the reviewer for his/her detailed feedback and concern regarding the general validity of the reactivity condition ($d\tau/dt^{\text{decay}} < 0$ values) and, more specifically, its *apparent* failure in the case of MeO-NAIP biomimetic rotor. Our short and direct answer is that, as also noticed by the reviewer, the validity of the $d\tau/dt^{\text{decay}} < 0$ condition is a function of the absolute magnitude of $d\tau/dt^{\text{decay}}$. In fact, our calculations indicate that the $d\tau/dt^{\text{decay}} < 0$ condition is only valid when the absolute value of $d\tau/dt^{\text{decay}}$ is above ca. 1 degrees/fs. Since, the distribution of $d\tau/dt^{\text{decay}}$ values in MeO-NAIP@MeOH displays significantly lower magnitudes with respect to PSB11@Rh then the $d\tau/dt^{\text{decay}} < 0$ condition is only partially valid in the biomimetic rotor. Thus, as the reviewer suspected we may have failed to make this clear in the text.

Notice that the data shown in the "population fraction vs hop times" plots in Figures 2b and 4b consistently reflect the trend shown in the histograms of Figures 2c and 4c. If $d\tau/dt^{\text{decay}} < 0$ condition for reactivity (dashed red line) was always valid (i.e. behaved like a sufficient condition), the dashed red curve would match the one representing the reactive trajectories (solid red line). However, because of its validity depend on the magnitude of the absolute value of $d\tau/dt^{\text{decay}}$, the dashed red line is higher than the population fraction of reactive trajectories.

Change/Remark 3. The following sentence has been added on page 5 of the main text in the hope to improve the clarity of the proposed reactivity theory.

“...It is shown that the resulting mechanistic interpretation leads to a generalization of theoretical framework based on the $d\tau/dt^{\text{decay}} < 0$ reactivity condition that also includes MeO-NAIP. Such a generalization implies that such condition is only valid when the absolute magnitude of $d\tau/dt^{\text{decay}}$ overcome a certain threshold. As discussed below this is qualitatively in line with the canonical Landau-Zener model...”

In the manuscript we formally express this result by introducing two, rather than one, conditions to be applied to each trajectory (molecule) of the molecular population:

- (i) the *necessary* condition ($d\tau/dt^{\text{decay}} < 0$) considered by the reviewer (being necessary but not sufficient, trajectories that satisfy such a condition may not be reactive), and
- (ii) a *sufficient* condition stating that condition “i” becomes sufficient only when a velocity threshold at decay is met (this second condition has been possibly overlooked by the reviewer).

If both conditions are considered, then our reactivity “theory” is valid for both Rh and MeO-NAIP. In fact, the sufficient condition “ii” is substantially always satisfied for Rh trajectories as almost all trajectories show $d\tau/dt^{\text{decay}}$ value overcome the established ca. 1 degree/fs threshold. This is not the case for MeO-NAIP where many trajectories decay with velocities below the threshold and, therefore, may or may not be reactive. Condition “ii” is thus critical for the correct interpretation of the histograms in Figures 2c and 4c. In fact, the “inconsistency” that the reviewer is highlighting (by applying only condition “i”) is central to our argument that, in general, the number of reactive trajectories produced by the simulations is not explained by a count of the trajectories with $d\tau/dt^{\text{decay}} < 0$ (i.e., the % of negative velocities at decay). Instead, the histograms indicate that a trajectory is predicted to be reactive only when $d\tau/dt^{\text{decay}} < 1$ degree/fs (i.e., the % of negative velocities satisfying condition “ii” at decay). Thus, in general, total number of reactive trajectories is a composite number comprising all trajectories satisfying condition “ii” plus ca. 50% of trajectories that decay with low absolute τ velocity.

Change/Remark 4. The following sentence on page 5 of the main text emphasizes an important point in the presented reactivity theory.

“...More specifically, i) we find that the $d\tau/dt^{\text{decay}} < 0$ condition is a *sufficient* condition only when the $d\tau/dt^{\text{decay}}$ amplitude is $\lesssim -1$ degree/fs...”

In the main manuscript we document the conclusion above rather extensively. For instance, our claim that $d\tau/dt^{\text{decay}} < 0$ is a *necessary condition* for reactivity is based on the statistics shown on Table 1. We analyze the population fraction of trajectories that decay to S_0 with $d\tau/dt^{\text{decay}} < 0$ velocities *within* the ‘reactive’ population fraction obtained in the simulation (i.e., the fraction actually achieving the photoproduct), and we find that in solution, 93% and 87% (in gas-phase 82% and 77%) of the retinal and biomimetic system respectively, decay with $d\tau/dt^{\text{decay}} < 0$ values. These are, qualitatively, similar percentages indicating that, statistically, the $d\tau/dt^{\text{decay}} < 0$ condition is satisfied for reactive trajectories. However, it does *not* indicate that a trajectory decaying with $d\tau/dt^{\text{decay}} < 0$ has a large probability of being reactive. The data of Table 1 point to the correct way to read the histograms in Figure 2c and 4c: from the total distribution of the reactive trajectories (red bars), the larger percentage is located on the negative velocities side of the histogram. But the reverse does not hold as not all trajectories with $d\tau/dt^{\text{decay}} < 0$ are comprised in the red bar as evident for the biomimetic system. Such a behavior is also supported by the percentages presented in Table 2 of the main text. Defining a “reactivity threshold” in terms of an absolute value of $d\tau/dt^{\text{decay}} > 1$, one can more confidently predict reactive trajectories using the $d\tau/dt^{\text{decay}} < 0$ condition that now appears valid for both the natural and biomimetic rotors. In contrast, trajectories with $-1 < d\tau/dt^{\text{decay}} < 1$ degree/fs may be reactive or unreactive. In the main manuscript we discuss how such *sufficient* condition for reactivity is intrinsically linked to the Landau-Zener model. Condition ii was not originally proposed for PSB11@Rh since for the natural rotor the majority of trajectories travel, at decay, with τ velocity above the threshold.

We propose the following changes to avoid any misunderstanding with our data presentation.

Change/Remark 5. The following periods on page 8 and 9 of the main text as well as Figures 2c and 4c and the corresponding legends have been updated for clarity. The incorrect inset in Figure 2c for MeO-NAIP@MeOH has been corrected. Notice that the scale of the population fraction of the insets in Figures 2c and 4c is now 0.0-0.2 to facilitate a comparative analysis.

“... $d\tau/dt^{\text{decay}} < 0$ is not a *sufficient condition for reactivity*. The data in Table 1 shows that 93% and 87% of the reactive trajectories of rPSB11@Rh and MeO-NAIP@MeOH are associated with $d\tau/dt^{\text{decay}} < 0$, while only 7% and 13% respectively are associated with $d\tau/dt^{\text{decay}} > 0$. If we now accept a lost 15% tolerance for the validity of the reactivity condition $d\tau/dt^{\text{decay}} < 0$, we can conclude that this is valid for both systems (the analog relationship $d\tau/dt^{\text{decay}} > 0 = \text{unreactive}$ is documented in Section IV of the Supplementary Information). Previous work on rPSB11@Rh, treated this as a sufficient condition implying that a trajectory with $d\tau/dt^{\text{decay}} < 0$ has a large probability of being reactive.^{15,20,21} However, while the simulation of the Rh dynamics, complies with a sufficient condition within a certain tolerance (Table 2 shows that 85% of the trajectories with $d\tau/dt^{\text{decay}} < 0$ are

indeed reactive), this is not true for MeO-NAIP@MeOH. In fact, in such system only 52% of the trajectories with $d\tau/dt^{\text{decay}} < 0$ are reactive. In other words, an almost equal proportion of reactive and unreactive trajectories has $d\tau/dt^{\text{decay}} < 0$, therefore $d\tau/dt^{\text{decay}} < 0$ cannot be, in general, a sufficient condition for reactivity...”

“...Indeed, in MeO-NAIP@MeOH, the whole population shows $d\tau/dt^{\text{decay}}$ with much lower absolute amplitudes (similarly having a small population fraction (13%) of reactive trajectories (see Table 1) with $d\tau/dt^{\text{decay}} > 0$)....”

“...In short, $d\tau/dt^{\text{decay}} > 0$ is in general valid only when the absolute value of $d\tau/dt^{\text{decay}}$ overcome a certain threshold...”

new Figure 2c

Change to the legend of Figure 2c.

“... Most reactive trajectories (ca. 90% in both cases) have $d\tau/dt^{\text{decay}} < 0$ while a minor number (ca. 10% in both cases) of trajectories with low positive amplitude $d\tau/dt^{\text{decay}} > 0$ are found to be reactive. The bins are 1.0 wide starting from -10.0. The insets show a finer distribution of the bins around the 0.0 value...”

new Figure 4c

Change to the legend of Figure 4c:

“...Most of reactive trajectories (ca. 80% in both cases) have $d\tau/dt^{\text{decay}} < 0$ while a minor number (ca. 20% in both cases) of trajectories with low positive amplitude $d\tau/dt^{\text{decay}} > 0$ are found to be reactive. The bins are 1.0 wide starting from -10.0. The insets show a finer distribution of the bins around the 0.0 value...”

Comment 2

The second fundamental argument to which I would have liked a convincing answer concerns the puckering promoter mode, which can strongly enhance the QY of the biomimetic system. The authors claimed that this important mode could be actually prevented in solvent by electronic reasons given by ‘counterion effects’. My personal doubt was that a reduction in the puckering mechanism could also be eventually induced by the steric hindrance of the solvent. Verification of this doubt seemed to me of some importance, because if the inhibition had steric reasons, the biomimetic system could not work in either polar or non-polar solvents ... possibly only in gas phase, which, from an application point of view, would be really limiting.

Response:

We considered the reviewer suggestion on the possible steric, rather than electronic, origin of the quenching or enhancement of the ring-puckering mode. We agree with the reviewer that a steric effect could, in principle, not be excluded. On the other hand, after having significantly expanded (see below) our investigation on the effect of DMSO (see also below) that has a higher viscosity relative to MeOH we conclude that it has not been possible to find evidence for a steric effect. This has now been reported in the main text of the revised manuscript.

Change/Remark 6. We have added the following period on page 23 of the main text.

“...The contribution of a steric, rather than electronic effect, to the quenching of the ring-inversion motion MeO-NAIP@MeOH cannot be excluded. In order, to partially investigate this process, we have run, using

exactly the same modeling methods, MeO-NAIP population dynamics in the solvent dimethyl sulphoxide (DMSO). DMSO is a polar aprotic solvent with higher (ca. four times) viscosity with respect to MeOH. Comparison between these solvent environments would thus allow to test (a) the effect of a higher viscosity as well as (b) lack of hydrogen bonding. As detailed in Section VI of the Supplementary Information, while MeO-NAIP@DMSO shows a qualitative behavior similar to the one in MeOH, the percentage of population undergoing ring-inversion in the pyrrolinium moiety is slightly increased most probably due to a change in S_1 electronic structure that may be hypothetically attributed to a decreased stabilization of the positive charge due to lack of hydrogen bonding. The increase in viscosity (i.e. a steric factor, presumably leading to a decreased ring-inversion motion) does not seem to be implicated in the described behavior...”

The authors tried to address this point by running one single dynamics in the aprotic solvent dimethyl sulfoxide (DMSO) following the same protocol already described for methanol solvent, starting from one randomly selected geometry of the sampled initial conditions. Then, they compared the evolution of the puckering mode on S_1 over time in the three different environments, i.e. gas phase, MeOH and DMSO. Since the trajectory in gas phase and in DMSO showed more similarities than that in MeOH, they valued this result optimistic over the hypothesis that it is a ‘counterion effect’ that prevents puckering promoter mode. But we clearly know that a single dynamic has no statistical meaning. A 0°K dynamics would have better served this cause. Alternatively, a quick, simple, and scientifically valid test could be to optimize the MeO-NAIP system in two specific structures, namely planar and distorted (puckering), in the three different environments respectively. Their corresponding energies clearly could tell us in which environment the puckering is most favored and then statistically more dynamically populated.

In conclusion, my two main doubts with respect to the proposed model are still pending. However, I believe that, if the model is robust, these nodes could be fixed and clearly explained.

Response:

We have dealt with the reviewer criticism at the best of our possibilities by computing 100, rather than one, MeO-NAIP trajectories in DMSO (MeO-NAIP@DMSO). In this way we have obtained statistical information to be compared, on an equal footing, with those obtained in isolated conditions and for MeO-NAIP@MeOH. We did not optimize S_1 MeO-NAIP in a planar and distorted (puckered) structure in the different environments, to avoid biases caused by selecting a specific solvent configuration when performing such a calculation as well as the effect of the torsional constraints to be necessarily imposed to avoid double bond twisting relaxation in the S_1 state during such a calculation. Remarkably, and contrary with the previously presented single trajectory analysis, we now find that DMSO and MeOH have similar effects on the population dynamics. *We are therefore very grateful to the reviewer for his/her criticism that has led us to avoid a qualitatively incorrect conclusion.*

The geometrical and charge analysis of the sets of MeO-NAIP trajectories in three different environments are now displayed in Figure S10 and S11 in the revised version of the Supplementary Information. Figure S10 shows that MeOH is most effective in quenching the population undergoing ring-inversion decreasing it from 66% in isolated condition (this is the percentage of the trajectories completing one oscillation) to a mere 26%. In DMSO the ring-inversion is quenched at a slightly lesser extent leading to a 34% of trajectories undergoing ring-inversion. The corresponding charge analysis in Figure S11 indicates that, similar to the MeOH environment, DMSO increases the stiffness of the pyrrolinium ring via an electronic rather than steric effect. However, this happens at a slightly lesser extent possibly resulting into an increased Φ^{iso} in this solvent. In contrast, we did not find evidence for an increased quenching effect due to the DMSO increase in viscosity.

Change/Remark 7. We have inserted the following sentence on page S16 in the Supplementary Information:

“...In the presence of a polar solvent, the S_1 electronic structure of the pyrroline ring contributes to stiffen the ring, significantly increasing the ring-inversion barrier and limiting the ring-puckering motion ...”

Change/Remark 8. We have updated and added new text to the following subsection including Figures S10 and S11 on pages S22-S25 of the Supplementary Information (the old Figure S11 is now Figure S12).

“...A further test was carried out to see if the Φ^{iso} inhibitory effect observed for MeO-NAIP in methanol is exclusively due to electronic reasons (as concluded above) or if additional effects (e.g., hydrogen bonding, steric hindrance, or others) brought up by a different environment is modulating the promoter motion. Accordingly, we constructed a model of MeO-NAIP in the aprotic polar solvent dimethyl sulfoxide (DMSO) following the same protocol already described for methanol solvent to evaluate the ring-puckering ρ mechanism in a different environment and extract some valuable conclusions.

We evaluated the ρ evolution along the S_1 population dynamic (500 fs) represented by 100 trajectories of the resulting MeO-NAIP@DMSO model and compared it with the population dynamics reported for isolated MeO-NAIP and MeO-NAIP@MeOH (see Figure S10). The results show that both polar solvents quench the ring-puckering motion with respect to isolated conditions. However, the population fraction that overpasses the planarity energy barrier and produced a ring oscillation between positive and negative ρ values is slightly higher in DMSO (34%) than MeOH (26%). Notice that the population fraction is selected based on trajectories that complete one oscillation within the

initial ca. 250 fs which also coincides with the S_1 lifetime (at least in isolated condition and MeOH). As discussed above, this quenched behavior in MeOH correlates with a reduction in quantum yield (Φ^{iso}), as it limits the fraction of the population capable of achieving significant ring distortion necessary to enhance the magnitude of $d\tau/dt$ velocity. The Φ^{iso} evaluation in DMSO is not discussed in the present study since this environment dramatically changes the excited state lifetime (beyond 500 fs) due to an interaction between bounded (S_2) and unbounded (S_1) electronic states limiting the number of MeO-NAIP@DMSO trajectories that decay to S_0 . The investigation of the DMSO has to be continued but due to the high computational cost could not be completed within the scope of the present study. Although we cannot correlate the higher percentage of the fraction conducting to oscillation with Φ^{iso} in DMSO, our MeO-NAIP@DMSO simulation demonstrates the potential for ring-inversion in each environment with very different features (volume and viscosity).

Figure S10. Evolution of the ρ angle across various environments. The top panel illustrates the full population dynamics, while the bottom panel highlights the percentage of sub-population fraction. achieving a complete oscillation within the initial ca. 250 fs coincides with the S_1 lifetime and the period of ρ . Black trajectories denote the population motion, and the yellow curves follow the average value.

Figure S11. Time progression of the average S_1 charge on the pyrrolinium moiety of MeO-NAIP in three different environments. (left) Charge evolution along the same trajectory set of Figure S10, top panels, and (right) Figure S10, bottom panels. In both panels it is evident the charge transfer nature of the S_1 state with respect to the S_0 state.

The analysis presented above is further validated by charge distribution studies of the pyrroline moiety on the S_1 state (see Figure S11). This confirms the dominant influence of electronic factors in governing the ring-puckering phenomena quenched in polar solvents. Collectively, our findings elucidate the complex interaction between molecular geometry, solvent interactions, and electronic structure, all of which are instrumental in shaping the photophysical properties of biomimetic rotors like MeO-NAIP...”

REVIEWERS' COMMENTS

Reviewer #2 (Remarks to the Author):

I would like to thank the authors for their comprehensive answers to the various questions I had asked for clarification. Accordingly, I would recommend this manuscript for publication in Nature Communications.